# Mirror Langevin Monte Carlo: the Case Under Isoperimetry

**Qijia Jiang**[*]
UT Austin
qjiang@austin.utexas.edu

## Abstract

Motivated by the connection between sampling and optimization, we study a mirror descent analogue of Langevin dynamics and analyze three different discretization schemes, giving nonasymptotic convergence rate under functional inequalities such as Log-Sobolev in the corresponding metric. Compared to the Euclidean setting, the result reveals intricate relationship between the underlying geometry and the target distribution and suggests that care might need to be taken in order for the discretized algorithm to achieve vanishing bias with diminishing stepsize for sampling from potentials under weaker smoothness/convexity regularity conditions.

## 1 Introduction

It has been widely recognized that optimization and sampling are deeply connected. One one hand, optimization can be viewed as performing sampling in the limit, and on the other, since the influential work of Jordan-Kinderlehrer-Otto [12], Langevin dynamics takes on the interpretation as performing deterministic optimization (gradient flow) in the space of probability measures. This profoundly shapes the way we view and understand traditional MCMC sampling algorithms, deviating from the Markov semigroup path. While huge amount of progress has been made on the optimization front in the past few decades, its sampling counterpart, finding far-reaching applications in Bayesian statistical inference and inverse problems, hasn't been fully explored to leverage the advancements offered by the optimization toolbox. In this paper, we draw inspiration from mirror descent [17] and ask the question if there's an analog of it that can adapt to geometries beyond the Euclidean case for Langevin diffusion, under isoperimetric inequalities such as Log-Sobolev for the target distribution, rather than Strong-Log-Concavity, where we recall the celebrated result of Bakry and Émery [3] prescribes that the latter implies the former.

### 1.1 Mirror Flow and Mirror Descent

In optimization, the extension to arbitrary geometry through the choice of a mirror map $\phi$ can often give better smoothness/strong convexity parameter dependence, or even handle cases where strong-convexity and Lipschitz gradient of $f$ do not hold in the Euclidean geometry. The continuous limit of mirror descent can be written as

$$dY_t = -\nabla f(X_t)dt, \quad X_t = \nabla\phi^*(Y_t) \tag{1}$$

which is equivalent to $dY_t/dt = d[\nabla\phi(X_t)]/dt = \nabla^2\phi(X_t)dX_t/dt = -\nabla f(X_t)$, therefore the mirror flow can be recast in the primal variable as $dX_t = -(\nabla^2\phi(X_t))^{-1}\nabla f(X_t)dt$, akin to natural gradient flow, which preconditions the update to adapt to local geometry. Equation (1) makes it clear that the mirror descent update

$$x_{k+1} = \nabla\phi^*(\nabla\phi(x_k) - h_{k+1}\nabla f(x_k)) = \arg\min_x \langle x, \nabla f(x_k)\rangle + h_{k+1}^{-1}D_\phi(x, x_k)$$

---

[*]Work done while at Stanford University.

is nothing more than the forward-discretized gradient descent in the dual $y$-space through mirror mapping. Here $D_\phi(x, x_k) = \phi(x) - \phi(x_k) - \nabla\phi(x_k)^\top(x - x_k) \geq 0$ is the Bregman divergence and $\phi^*$ is the Fenchel conjugate of $\phi$. We assume that the mirror map $\phi: \mathbb{R}^d \to \mathbb{R}$ is of Legendre type[2] and strictly convex throughout. Common choices include $\phi(x) = \|x\|_2^2/2$, which reduces to classical gradient descent as $x_{k+1} = x_k - h_{k+1}\nabla f(x_k)$, and $\phi(x) = -\sum_i x_i \log(x_i)$, which gives multiplicative weight update. In the special case when $\phi = f$, one readily recovers Newton's method.

## 1.2  Mirror Langevin Dynamics and Mirror Langevin Monte Carlo

For sampling, we consider the Mirror Langevin stochastic differential equation (SDE) where for $Y_t = \nabla\phi(X_t)$, and target distribution $\pi = e^{-f}$,

$$dY_t = -\nabla f(\nabla\phi^*(Y_t))dt + \sqrt{2[\nabla^2\phi^*(Y_t)]^{-1}}dW_t \qquad (2)$$

for $W_t$ the standard Brownian motion in $\mathbb{R}^d$. If $\phi$ is three-times-differentiable, it is equivalent to

$$dX_t = \left(-[\nabla^2\phi(X_t)]^{-1}\left[\mathrm{Tr}\left(\nabla^3\phi(X_t)[\nabla^2\phi(X_t)]^{-1}\right) - \nabla f(X_t)\right]\right)dt + \sqrt{2[\nabla^2\phi(X_t)]^{-1}}dW_t \qquad (3)$$

and the corresponding Euler-Maruyama (EM) discretized version (in dual $y$-space) becomes

$$x_{k+1} = \nabla\phi^*\left(\nabla\phi(x_k) - h_{k+1}\nabla f(x_k) + \sqrt{2h_{k+1}[\nabla^2\phi(x_k)]} \cdot z_{k+1}\right) \qquad (4)$$

for $h_{k+1}$ the stepsize and $z_{k+1} \in \mathbb{R}^d$ an independent standard Gaussian random vector, where we used $\nabla^2\phi^*(Y_t) = [\nabla^2\phi(X_t)]^{-1}$. No particular warm start is assumed for initialization. It is worth noting that while the continuous dynamics in the primal $X$-space involves the 3rd-order derivative tensor, the implementation of (4) only requires access to a gradient oracle for $f$. For $\phi(x) = \|x\|_2^2/2$, one recovers the classical (overdamped) Langevin dynamics, whose Euler-Maruyama discretization is ULA: $x_{k+1} = x_k - h_{k+1}\nabla f(x_k) + \sqrt{2h_{k+1}} \cdot z_{k+1}$. In the case when there is no closed-form expression, the inversion $\nabla\phi^*$ can be solved numerically (and therefore approximately) using $\nabla\phi^*(z) = \arg\max_x\{z^\top x - \phi(x)\}$, which is a convex optimization problem. A derivation for the equivalence between (2) and (3), along with the fact that (3) has $\pi = e^{-f}$ as the stationary distribution are given in Appendix A.

It is evident (and reminiscent of the classical Langevin algorithm) that the discretized algorithm will converge to a biased limit $\pi^h$ under mild regularity conditions. A Metropolis-Hastings step can be applied on top to correct for the bias but we focus on the unadjusted case in this paper. The main question we aim to address in this paper is – what is the non-asymptotic rate of convergence for (2) using different discretization schemes, under functional inequalities such as Log-Sobolev Inequality (LSI), which encompasses broader classes of distributions compared to the more restrictive and well-studied Strong-Log-Concavity (SLC) setting. In particular, it is known that LSI is preserved under bounded perturbation [10] and Lipschitz mapping, which capture cases when $f$ is far from convex, e.g., multi-modal such as Gaussian mixtures.

## 2  Related Work

Various discretizations of underdamped, overdamped Langevin dynamics in the Euclidean setting under LSI [14, 23, 15] and SLC [8, 5, 7, 19, 16] are the main focus of a series of developments, for which non-asymptotic error bounds are established for several metrics including KL, Wasserstein and TV distance. Convergence of discretized algorithm under LSI, in general, introduces considerably more challenges, as commonly used synchronous/reflection coupling techniques do not apply.

For Langevin dynamics working under non-Euclidean geometry, an earlier proposal was made in [11] where an algorithm was designed to converge to $(\nabla\phi)_\#\pi$, being essentially a change of measure from the classical Langevin dynamics. Crucially, their dynamics is different in that the diffusion term isn't scaled as the one we consider to take into account the Riemannian metric structure induced by $\phi$. The study of dynamics (2) was initiated in [24] under a relaxed-SLC assumption, where the authors show convergence of (4) to a Wasserstein ball with non-vanishing bias. [1] studied under a similar relaxed-SLC assumption, but with a different discretization scheme as opposed to (4),

---

[2]so that $\nabla\phi$ is invertible and $\nabla\phi^* = (\nabla\phi)^{-1}$ is a single-valued mapping that makes sense for (1)

where the bias decreases to zero with diminishing stepsize. Closer to our work is [6], in which the authors investigated convergence of the continuous process (2) under functional inequalities resembling the ones we consider and focused on $\phi = f$, leaving the analysis of discretized algorithm for future work. In this case one does not need to resort to smoothness assumption etc. for deriving contraction/handling bias issues – it is only when studying stable discretization schemes that they become necessary (and slows down convergence from exponential to polynomial).

Recent work of [13] leverages the mean-square framework, which heavily exploits the contraction property of the dynamics, to show that under a modified self-concordance property, algorithm (4) in fact converges without asymptotic bias. This particular property is something we do not have here, therefore it's likely new ideas are needed if one were to improve the analysis of (4) in this setting.

## 3  Notation and Assumptions

**Notation**   We use $\nabla \cdot v(x) = \sum_i \frac{\partial v_i(x)}{\partial x_i} \in \mathbb{R}$ to denote the divergence operator for a vector field $v : \mathbb{R}^d \to \mathbb{R}^d$, and for a matrix-valued function $G : \mathbb{R}^d \to \mathbb{R}^{d \times d}$, we let $\nabla \cdot G(x) \in \mathbb{R}^d$ with the $i$-th element as $\sum_j \frac{\partial G(x)_{ij}}{\partial x_j}$. We denote the Laplacian operator as $\Delta$ where $\Delta \phi(x) = \mathrm{Tr}(\nabla^2 \phi(x))$. Moreover, let $\langle \nabla^2, G(x) \rangle = \nabla \cdot (\nabla \cdot G(x)) = \sum_{i,j} \frac{\partial^2 G_{ij}(x)}{\partial x_i \partial x_j} \in \mathbb{R}$. The norm induced by a positive definite $G$ is defined as $\|z\|_G^2 := z^\top G z$. Wasserstein-$p$ distance is defined as $W_p(\rho, \pi) = \inf_{x \sim \rho, x' \sim \pi} \mathbb{E}[\|x - x'\|^p]^{1/p}$ for $p \geq 1$. From Monge-Kantorvich's duality $W_1(\rho, \pi) = \sup_{\|f\|_{lip} \leq 1} \mathbb{E}_\rho[f] - \mathbb{E}_\pi[f]$ and monotonicity property $W_1(\rho, \pi) \leq W_2(\rho, \pi)$, bound on Wasserstein-2 metric captures many functions of potential interest. We define the pushforward measure $\bar{\rho} = \nabla \phi_{\#} \rho$ as $\bar{\rho}(\mathcal{B}) = \rho(\nabla \phi^{-1}(\mathcal{B}))$ for every borel set $\mathcal{B}$.

We state the assumptions that we will make below, some of which we will relax later.

**Assumption 1** ($\zeta$-Self-Concordance). *There exists a constant $\zeta \geq 0$ such that the conjugate mirror map $\phi^*$ satisfies that $\forall y, u, s, v$,*

$$|\nabla^3 \phi^*(y)[u, s, v]| \leq 2\zeta \cdot (u^\top \nabla^2 \phi^*(y)u)^{1/2}(s^\top \nabla^2 \phi^*(y)s)^{1/2}(v^\top \nabla^2 \phi^*(y)v)^{1/2}.$$

*Moreover, this property is preserved under Fenchel conjugation (with the same parameter), affine transformation and summation [18].*

Many natural barrier and entropy functions (e.g., log-barrier) satisfy such self-concordance property. This also guarantees solution of the continuous dynamics (2), cf. Appendix A of [24]. One can show that this affine-invariant condition implies a form of Hessian stability as $M^{-1}\nabla^2 \phi(x) \preceq \nabla^2 \phi(x') \preceq M\nabla^2 \phi(x)$ for some $M \geq 1$ (cf. Lemma 4), entailing that the underlying geometry isn't rapidly changing. This form of self-concordance also appears in interior point method in optimization and previous work on Mirror Langevin [1] and suggests that locally the function behaves like a quadratic.

**Assumption 2** ($\beta$-Mirror-Log-Sobolev). *The target distribution $\pi$ satisfies Mirror LSI with constant $\beta$ w.r.t a given mirror map $\phi$, i.e., for every locally lipschitz function $g$, it holds that $\pi$ satisfies*

$$\frac{2}{\beta} \int \|\nabla g(x)\|_{[\nabla^2 \phi(x)]^{-1}}^2 d\pi \geq \int g(x)^2 \log g(x)^2 d\pi - \left(\int g(x)^2 d\pi\right) \log \left(\int g(x)^2 d\pi\right) \quad (5)$$

*taking $g(x) = \sqrt{d\rho(x)/d\pi(x)}$ one gets that for all $\rho$,*

$$H_\pi(\rho) := \int \rho(x) \log \frac{\rho(x)}{\pi(x)} dx \leq \frac{1}{2\beta} \int \rho(x) \left\|\nabla \log \frac{\rho(x)}{\pi(x)}\right\|_{[\nabla^2 \phi(x)]^{-1}}^2 dx =: \frac{1}{2\beta} J_\pi^\phi(\rho). \quad (6)$$

*This is the gradient-domination condition for KL-divergence in the Wasserstein metric, where the (weighted) Fisher information on the RHS is the squared norm of the gradient for KL divergence.*

LSI is an isoperimetric inequality that implies concentration for the distribution (subgaussian tails on Lipschitz functions) and plays an important role in many results in probability theory. A discussion of its implication in our context is further expanded in Section 4.1. We note that both [1] and [24] require relative $\mu$-strong convexity of $f$ w.r.t $\phi$, which means $\nabla^2 f \succeq \mu\nabla^2 \phi \succ 0$. Therefore this assumption based on LSI allows us to move away from convex potentials.

The following two recover the familiar smoothness and Lipschitz condition when $\phi(x) = \|x\|_2^2/2$.

**Assumption 3** (L-Relative Lipschitz). *For all $x$, it holds that $f : \mathbb{R}^d \mapsto \mathbb{R}$ is differentiable with*

$$\|\nabla f(x)\|_{[\nabla^2 \phi(x)]^{-1}} \leq L.\tag{7}$$

**Assumption 4** ($\gamma$-Relative Smooth). *For all $x, x' \in dom(\phi)$,*

$$\|[\nabla^2\phi(x)]^{-1}\nabla f(x) - [\nabla^2\phi(x')]^{-1}\nabla f(x')\|_{\nabla^2\phi(x')} \leq \gamma \cdot \|\nabla\phi(x) - \nabla\phi(x')\|_{[\nabla^2\phi(x')]^{-1}}.\tag{8}$$

One could show that (8) is slightly stronger than assuming $\nabla^2 \tilde{f}(y) \preceq \gamma \nabla^2 \phi^*(y)$ for $\tilde{f}(y) = f(\nabla\phi^*(y))$ or that $\tilde{f}(y)$ is smooth, i.e., $\nabla^2 \tilde{f}(y) \preceq c \cdot I$ (cf. Lemma 7). When $\phi(x) = \|Ax\|_2^2/2$, condition (8) reduces to $\|\nabla f(x) - \nabla f(x')\|_{(A^\top A)^{-1}} \leq \gamma \cdot \|x - x'\|_{A^\top A}$, for which the dual norm on the two sides makes sense as a generalization, whereas one could check that the relative smoothness as defined in [24, 1] do not admit such a natural interpretation. We will comment more on this assumption in later sections on discretization. Condition (7) is also common in previous works [1].

**Assumption 5** ($\alpha$-Strongly Convex). *The mirror map $\phi$ is three-times differentiable, and let $\alpha := \lambda_{\min}(\nabla^2\phi) > 0$.*

All best known results under LSI [14, 23] in the Euclidean setting assume third order smoothness on $f$ (i.e., Lipschitz Hessian) and Lipschitz gradient (i.e., $-L \cdot I \preceq \nabla^2 f \preceq L \cdot I$), sometimes with an additional disspativity assumption [15], whereas we only require a weak notion of smoothness on $f \circ \nabla\phi^*$. The study of discretized sampling algorithms for non-smooth and non-convex potentials, to the best of our knowledge, is a road less traversed.

## 4 Convergence Analysis

### 4.1 Continuous Time Process: Mirror LSI

One can get using LSI (and LSI alone) the exponential convergence in KL divergence for the continuous time process, which is a manifestation of convergence under Polyak-Łojasiewicz (PL) inequality from optimization. In optimization, steepest descent gradient flow curve $y_t^* \in \mathbb{R}^d$ is defined as

$$\min_{y_t} \langle \nabla f(y_t), \dot{y}_t \rangle + \frac{1}{2}\|\dot{y}_t\|^2$$

which implies $\dot{y}_t^* = -\nabla f(y_t^*)$ and $df(y_t^*)/dt = -\|\nabla f(y_t^*)\|^2$, from which $m$-gradient dominance condition on $f$ (weaker than strong convexity)

$$f(y_t^*) - \min_y f(y) \leq \frac{1}{m}\|\nabla f(y_t^*)\|^2$$

gives exponential convergence for the objective function $f$ as $d(f(y_t^*) - \min_y f(y))/dt \leq -m(f(y_t^*) - \min_y f(y))$. In sampling within the space of probability measures, the objective function(al) is replaced by $H_\pi(\rho_t)$, and the seminal work of [12] (see also later treatments by [2, 22]) show that the density $\rho_t \in \mathcal{P}(\mathbb{R}^d)$ followed by the (overdamped) Langevin dynamics satisfies

$$\dot{\rho}_t = \arg\min_{v_t \text{ tangent to } \rho_t} \mathbb{E}_{x\sim\rho_t}\left[\langle \nabla_{W_2} H_\pi(\rho_t)(x), v_t(x) \rangle\right] + \frac{1}{2}\mathbb{E}_{x\sim\rho_t}[\|v_t(x)\|_2^2]$$

where $\nabla_{W_2} H_\pi(\rho_t)(x) = \nabla_x \frac{\partial H_\pi(\rho_t)}{\partial\rho_t}(x)$ is the Wasserstein-2 gradient of KL divergence at $\rho_t$. Therefore

$$\frac{dH_\pi(\rho_t^*)}{dt} = -\mathbb{E}_{x\sim\rho_t^*}\left[\left\|\nabla_x \frac{\partial H_\pi(\rho_t^*)}{\partial\rho_t}(x)\right\|_2^2\right] = -\mathbb{E}_{x\sim\rho_t^*}\left[\left\|\nabla_x\left(1 + \log\left(\frac{\rho_t^*(x)}{\pi(x)}\right)\right)\right\|_2^2\right],$$

where for the second step we used $\frac{\partial H_\pi(\rho_t)}{\partial\rho_t} = 1 + \log(\frac{\rho_t}{\pi})$ [2]. The RHS becomes the negative of the Fisher information $J_\pi^\phi(\rho)$ therefore LSI (6) reduces to gradient dominance condition for KL divergence (in this case with $\nabla^2\phi = I$ for classical Langevin dynamics), from which one can derive exponential convergence rate to the target distribution $\pi$ as carried out in the optimization land sans bias. This casts Langevin diffusion as precisely gradient flow w.r.t KL divergence in Wasserstein metric in the space of probability measures, from which LSI (weaker than SLC) suffices for convergence. We show in the lemma below that there is an extension of this result for Mirror-LSI that gives exponential convergence for the continuous process (3).

**Proposition 1** (Convergence under Mirror-LSI). *Along the dynamics of (3), we have that $H_\pi(\rho_t) \leq e^{-2\beta t} H_\pi(\rho_0)$ under Assumption 2.*

*Proof.* Using the PDE (22), which describes the density followed by (3) and the integration by parts formula $\int \langle \nabla \phi(x), v(x) \rangle dx = -\int \phi(x) \nabla \cdot v(x) dx$,

$$
\begin{aligned}
\frac{d}{dt} H_\pi(\rho_t) &= \int \frac{d\rho_t}{dt} \log \frac{\rho_t}{\pi} dx + \int \pi \frac{1}{\pi} \frac{d\rho_t}{dt} dx \\
&= \int \nabla \cdot \left( \rho_t [\nabla^2 \phi]^{-1} \nabla \log \frac{\rho_t}{\pi} \right) \log \frac{\rho_t}{\pi} dx + \frac{d}{dt} \int \rho_t dx \\
&= -\int \rho_t \left\| \nabla \log \frac{\rho_t}{\pi} \right\|_{[\nabla^2 \phi]^{-1}}^2 dx + 0 \\
&\leq -2\beta \cdot H_\pi(\rho_t)
\end{aligned}
$$

where we used Mirror LSI in the last step. This implies from Grönwall's inequality that $H_\pi(\rho_t) \leq e^{-2\beta t} H_\pi(\rho_0)$. □

For $\pi$ that satisfies Mirror LSI with constant $\beta$, thanks to the strong convexity of $\nabla^2 \phi \succeq \alpha I$, it also satisfies Talagrand's inequality [21] with parameter $\beta \cdot \alpha$, which means the Wasserstein-2 distance is upper bounded by KL divergence as $\frac{\alpha\beta}{2} W_2(\rho, \pi)^2 \leq H_\pi(\rho)$ for any $\rho$. Such transportation-cost inequality has the interpretation of quadratic growth in the iterate space. Therefore Mirror LSI has the additional nice property of allowing us to translate guarantee in the objective value (i.e., KL divergence) to iterate space (that involves optimal coupling between iterates $x \sim \rho, x' \sim \pi$).

**Stability** One can show that Mirror-LSI, similar to its Euclidean counterpart, is stable under bounded perturbation. Therefore if the potential function $f$ of interest is not exactly relative-smooth w.r.t to a "nice" mirror map, it suffices for it to be close to one, in the sense made precise in Appendix B. This is not something one can hope for with strong-convexity assumption, where perturbation from convex function usually breaks the assumption. This could be especially useful when the potential takes a composite form of $f + g$, where one part is smooth and the other part isn't. It is also known that operations such as convolution with Gaussian (or other density that satisfy LSI) preserves LSI as well [4], which offers the option of smoothing to perform approximate sampling from a nicer proxy potential $\tilde{f}$. All these reasons make target measure satisfying Mirror-LSI appealing to study compared to the two previous works [1, 24] studying mirror Langevin for $\nabla^2 f \succ 0$.

## 4.2 EM-Discretized Process: Interpolation with Weighted Dynamics

For analyzing the EM discretization, we build upon the idea initiated in [23] and view the discretized mirror Langevin Monte Carlo (4) as following a *weighted* Langevin dynamics. It is clear that $y_{k+1} = \nabla \phi(x_{k+1})$ is the value at time $t = h_{k+1}$ of the stochastic process

$$ Y_t = Y_0 - t \cdot \nabla f(\nabla \phi^*(Y_0)) + \sqrt{2[\nabla^2 \phi^*(Y_0)]^{-1}} W_t $$

starting from $Y_0 = y_k$, or written in differential equation form,

$$ dY_t = -\nabla f(\nabla \phi^*(Y_0)) dt + \sqrt{2[\nabla^2 \phi^*(Y_0)]^{-1}} dW_t. \tag{9} $$

Through the mapping $X_t = \nabla \phi^*(Y_t)$, one can study the corresponding dynamics in $X$-space, which evolves following a *weighted* Langevin dynamics with shifted drift $\hat{\mu}$ as shown in Lemma 1 (that is responsible for convergence to a biased limit $\pi^h \neq \pi$), i.e.,

$$ dX_t = (\nabla \cdot G(X_t) - G(X_t) \nabla f(X_t) + \hat{\mu}) dt + \sqrt{2G(X_t)} dW_t. \tag{10} $$

**Lemma 1** (Shifted Drift and Covariance). *For the dynamics written in (10) following (9), we have $G(X_t) = [\nabla^2 \phi(X_t)]^{-1} \nabla^2 \phi(X_0) [\nabla^2 \phi(X_t)]^{-1} \succ 0$ and*

$$
\begin{aligned}
\hat{\mu} = &-[\nabla^2 \phi(X_t)]^{-1} \nabla f(X_0) - [\nabla^2 \phi(X_t)]^{-1} \operatorname{Tr}\left( \nabla^3 \phi(X_t) [\nabla^2 \phi(X_t)]^{-1} \nabla^2 \phi(X_0) [\nabla^2 \phi(X_t)]^{-1} \right) \\
&- \nabla \cdot \left( [\nabla^2 \phi(X_t)]^{-1} \nabla^2 \phi(X_0) [\nabla^2 \phi(X_t)]^{-1} \right) + [\nabla^2 \phi(X_t)]^{-1} \nabla^2 \phi(X_0) [\nabla^2 \phi(X_t)]^{-1} \nabla f(X_t).
\end{aligned}
$$

Having established this, a combination of Mirror LSI and careful bounding of the discretization error using self-concordance and smoothness properties can be brought together to derive the per-iteration progress. Deatiled proofs for this section can be found in Appendix C.

**Proposition 2** (Progress in One Step of EM Discretization). *In one iteration of Algorithm (4) with $x_k \sim \rho_0, x_{k+1} \sim \rho_h$, under Assumption 1- 5, define $D := \max_{u,v} \|\nabla\phi(u) - \nabla\phi(v)\|_2$, we have for stepsize $h \leq \min(1/2\zeta L, 1/16\zeta^2 d, D/\sqrt{\alpha}L, D^2/4\alpha d, M/6\beta)$,*

$$H_\pi(\rho_h) \leq e^{-\frac{3\beta}{2M}h}H_\pi(\rho_0) + 24M^2\gamma^2 dh^2 + 16M\zeta^2 d^2\eta_h^2 h,$$

*where we denote $M = \exp(2\zeta D/\sqrt{\alpha})$ and $\eta_h^2 = (1 - \exp(-1/16\zeta^2 h)) \cdot (1 - \zeta(hL + 2\sqrt{hd}))^{-4} + \exp(-1/16\zeta^2 h) \cdot M^2$. We use the convention $M = 1$ when $\zeta = 0$ and $D = \infty$.*

Picking appropriate stepsize gives the following result – perhaps surprisingly, this particular analysis suggests that the simplest EM discretization exhibits an irreducible bias, and similar observation was also made in [24] where the authors showed convergence to a Wasserstein ball with explicit radius, even for diminishing stepsize.

**Theorem 1** (Convergence Guarantee for EM). *Under Assumption 1-5, picking stepsize $h \leq \min(1/2\zeta L, 1/16\zeta^2 d, D/\sqrt{\alpha}L, D^2/4\alpha d, M/6\beta, \delta\beta/44M^3\gamma^2 d)$, after $k \geq \tilde{\Omega}(M^4\gamma^2 d/\beta^2\delta)$ iterations of Algorithm (4), we have for $x_k \sim \rho_k$ that $H_\pi(\rho_k) \leq \delta + R_h$, where the nonvanishing bias $R_h = \mathcal{O}(M^2\zeta^2 d^2/\beta)$. In the above, $D = \max_{u,v} \|\nabla\phi(u) - \nabla\phi(v)\|_2$ and $M = \exp(2\zeta D/\sqrt{\alpha})$.*

*Proof.* Iterating the inequality in Proposition 2 for $k$ iterations,

$$H_\pi(\rho_k) \leq e^{-\frac{3\beta}{2M}hk}H_\pi(\rho_0) + \frac{24M^2\gamma^2 dh^2}{1 - e^{-\frac{3\beta}{2M}h}} + \frac{16M\zeta^2 d^2\eta_h^2 h}{1 - e^{-\frac{3\beta}{2M}h}}$$

$$\leq e^{-\frac{3\beta}{2M}hk}H_\pi(\rho_0) + \frac{22M^3\gamma^2 dh}{\beta} + \frac{15M^2\zeta^2 d^2\eta_h^2}{\beta}$$

where we used $1 - e^{-a} \geq 3a/4$ for $a \in (0, 1/4]$. Now using Lemma 6 for initialization, picking the assumed stepsize, after $k \geq \tilde{\Omega}(M/\beta h) \geq \tilde{\Omega}(M^4\gamma^2 d/\beta^2\delta)$ iterations, we have $H_\pi(\rho_k) \leq \delta + R_h$. As long as $\nabla^2\phi$ is not constant (therefore $\zeta \neq 0$, recall $M \geq 1$), $R_h := 15M^2\zeta^2 d^2\eta_h^2\beta^{-1} \neq 0$ as $h \to 0$ and the asymptotic bias $R_h$ scale as $\mathcal{O}(M^2\zeta^2 d^2/\beta)$ since $\eta_h^2 \to 1$ as $h \to 0$. $\square$

*Remark.* This convergence rate for KL divergence is stronger than [24] with their guarantee in Wasserstein distance. Additionally, by Pinsker's inequality, TV distance is upper bounded by KL divergence therefore one could also get an analogous guarantee in that metric. Variations on the argument will likely generalize to other metrics such as $\chi^2$ and Rényi divergence, with other appropriate (mirror-version of) functional inequalities such as Poincaré [6, 20].

In the case when $\phi(x) = \|Ax\|^2/2$, Theorem 1 gives no asymptotic bias, as we have $M = 1$ and $\zeta = 0$. Moreover, the assumption in this case says $f$ is smooth in $\|\cdot\|_{A^\top A}$ although not necessarily convex, and our algorithm gives an update of the form $x_{k+1} = x_k - h_{k+1}(A^\top A)^{-1}\nabla f(x_k) + \sqrt{2h_{k+1}(A^\top A)^{-1}}z_{k+1}$, which coincides with the classical Langevin performed on the function $g(\tilde{x}) = f(A^{-1}\tilde{x})$ for $x = A^{-1}\tilde{x}$. One could check that $g$ is smooth $-\gamma \cdot I \preceq \nabla^2 g \preceq \gamma \cdot I$ and satisfies the classical LSI, which means that $f$ is globally nice w.r.t a fixed geometry, after a change of basis. If applying the Euclidean result of [20] on $g(\cdot)$, Theorem 1 recover the same $\tilde{\mathcal{O}}(d/\delta)$ complexity, which is the best-known rate without third-order smoothness assumption on $f$.

A closer look at the theorem points out that $M$ plays a prominent role in the rate, where $M$ is effectively the Hessian stability parameter (cf. Lemma 4). However, this dependence on $M$ necessarily means that $\phi$ needs to be smooth on its domain (and $M$ will be roughly the condition number of $\phi$). Although this extends beyond the Euclidean case when $\phi$ needs to be constant – hence allowing for slowly-changing geometry, it is still far from satisfying. An important motivation for relaxing smoothness assumption is in constrained sampling (e.g., uniform sampling from a convex body), where one typically would pick $\phi$ to be a self-concordant barrier function that blows up on the boundary as a proxy for the nonsmooth constraint for approximate sampling. Such functions do not have a bounded $M$. This, in addition to the non-vanishing bias, all suggest that EM discretization, natural as it may seem, although could improve particular parameter dependence compared to the Euclidean counterpart (as illustrated above), might not fully benefit from the use of a mirror map.

## 4.3 Alternative Forward Discretization Scheme

In practical applications, it is often the case that the cost of evaluating $\nabla\phi$ is considerably cheaper than the cost of computing $\nabla f$, which could involve a finite sum over a large number of data points. Taking hints from this observation, in [1], a slightly modified mirror Langevin algorithm was considered where at iteration $k$, with step size $\eta$

$$x_{k+1/2} = \arg\min_v \eta\nabla f(x_k)^\top v + D_\phi(v, x_k) = \nabla\phi^*(\nabla\phi(x_k) - \eta\nabla f(x_k)) \tag{11}$$

$$\text{solve } dy_t = \sqrt{2[\nabla^2\phi^*(y_t)]^{-1}}dW_t \ \text{ for } \ y_0 = \nabla\phi(x_{k+1/2}) \tag{12}$$

$$x_{k+1} = \nabla\phi^*(y_\eta) \tag{13}$$

The oracle complexity (i.e., number of queries for $\nabla f$) of the above algorithm is the same as the one in (4), but aiming at a higher accuracy implementation for the diffusion part involving $\phi$. The inner step (12) can be implemented approximately using e.g., Euler-Maruyama. Integrating both sides of (12), it is not hard to see that $y_{k+1} = \nabla\phi(x_{k+1})$ is the value at time $t = \eta$ of the continuous process

$$Y_t = Y_0 - t\nabla f(X_0) + \sqrt{2}\int_0^t [\nabla^2\phi^*(Y_s)]^{-1/2}dW_s \tag{14}$$

given $X_0 = \nabla\phi^*(Y_0) = x_k$ from the previous iteration. Written in differential form,

$$dY_t = -\nabla f(\nabla\phi^*(Y_0))dt + \sqrt{2[\nabla^2\phi^*(Y_t)]^{-1}}dW_t \,. \tag{15}$$

Compared to (9), the difference is in the second term where we traded $Y_0$ for $Y_t$, therefore this formulation amounts to discretizing the objective but not the geometry and will turn out to be crucial for removing the asymptotic bias. This is also in line with the observation from optimization [9], from which the authors argue that $\dot{x}_t = -\nabla^2\phi(x_t)^{-1}\nabla f(x_{\lfloor t\rfloor})$ gives a more "faithful" discretization compared to $\dot{x}_t = -\nabla^2\phi(x_{\lfloor t\rfloor})^{-1}\nabla f(x_{\lfloor t\rfloor})$. Indeed for the process (15), one gets in the $X$-space another weighted Langevin dynamics (10) with $G = [\nabla^2\phi(X_t)]^{-1}$ and $\hat{\mu} = [\nabla^2\phi(X_t)]^{-1}(\nabla f(X_t) - \nabla f(X_0))$, the discretization error of the gradient in the local $\phi$ metric.

It turns out that this algorithm based on splitting the deterministic and stochastic part of the SDE works with a weaker notion of smoothness assumption as well. In particular, this definition of relative smoothness only involves the local metric $\nabla^2\phi$ at a single point, whereas the previous Assumption 4 requires Lipschitz gradient across different metrics $\nabla^2\phi$, which might be unavoidable if one is discretizing the geometry as well.

**Assumption 6** (Weaker $\gamma$-Relative Smooth). *For all $x, x' \in dom(\phi)$, it holds that*

$$\|\nabla f(x) - \nabla f(x')\|_{[\nabla^2\phi(x')]^{-1}} \leq \gamma \cdot \|\nabla\phi(x) - \nabla\phi(x')\|_{[\nabla^2\phi(x')]^{-1}} \,.$$

When $\phi(x) = \|Ax\|_2^2/2$, Assumption 6 reduces to $\|\nabla f(x) - \nabla f(x')\|_{(A^\top A)^{-1}} \leq \gamma \cdot \|x - x'\|_{A^\top A}$; and when $\phi = f$ we always have $\gamma = 1$ (not the case for Assumption 4). We have the following result for the forward-discretized Mirror Langevin algorithm. Proofs for this section can be found in Appendix D.

**Proposition 3** (Convergence Guarantee for Forward Discretization). *For the Algorithm in (11)-(13), under Assumption 1,3,5,6, let $M = \exp(2\zeta D/\sqrt{\alpha})$ and $D = \max_{u,v}\|\nabla\phi(u) - \nabla\phi(v)\|_2$, picking stepsize $h \leq \min(1/2\zeta L, 1/16\zeta^2 d, D/\sqrt{\alpha}L, D^2/4\alpha d, 1/6\beta, \delta\beta/100M\gamma^2 d)$, after $k \geq \tilde{\Omega}(M\gamma^2 d/\beta^2\delta)$ iterations, we have $H_\pi(\rho_k) \leq \delta$.*

While it is reassuring that the algorithm has vanishing bias with diminishing stepsize for any self-concordant mirror map $\phi$ (at a higher cost of computation for each step), we still see the appearance of $M$ in the rate, which as we discussed earlier, substantially limit the use case for handling weakly smooth potentials as motivation for the introduction of a mirror map. To give a concrete example, for logistic regression, one might be interested in sampling from a posterior $\pi(\theta) \propto \lambda(\theta) \cdot \exp[\sum_i y_i\theta^\top x_i - \log(1 + \exp(\theta^\top x_i))]$, where $\lambda(\theta)$ could be a constrained prior such as uniform on $\ell_\infty$ ball $[-1, 1]^d$. In such a setting, one natural choice is to pick a mirror map which is a self-concordant barrier for the constraint set to enforce the constraint on the drawn samples, e.g., $\phi(\theta) = \sum_i \log((1 - \theta_i)^{-1}) + \log((1 + \theta_i)^{-1})$ with $dom(\phi) = (-1, 1)^d$. One could check that with the potential of interest as $f(\theta) = \sum_i -y_i\theta^\top x_i + \log(1 + \exp(\theta^\top x_i))$, it does not have a bounded $M$.

## 4.4 Alternative Backward Discretization Scheme

Backward discretization is known to be more stable compared to forward discretization in optimization and it is also known to give the best rate under LSI for Langevin diffusion in the Euclidean setting with weaker assumptions [23], albeit at a higher cost of solving a proximal step $x_{k+1} = x_k - \eta \nabla f(x_{k+1}) = \arg\min_x f(x) + (2\eta)^{-1}\|x - x_k\|_2^2$ compared to $x_{k+1} = x_k - \eta\nabla f(x_k)$ at each step.

It is relatively straightforward to see that a backward discretization for the dynamics using the same philosophy as Section 4.3 can be implemented with step size $\eta$ as (assuming $\eta f + \phi$ is convex - guaranteed if $\eta$ small enough)

$$\text{solve } dy_t = \sqrt{2[\nabla^2\phi^*(y_t)]^{-1}}dW_t \text{ for } y_0 = \nabla\phi(x_k) \tag{16}$$

$$x_{k+1} = \arg\min_v \eta f(v) + \phi(v) - y_\eta^\top v \Leftrightarrow \eta\nabla f(x_{k+1}) + \nabla\phi(x_{k+1}) - y_\eta = 0 \tag{17}$$

and $y_{k+1} = \nabla\phi(x_{k+1})$ is the value at time $t = \eta$ of the continuous process

$$Y_t = Y_0 - t\nabla f(\nabla\phi^*(Y_t)) + \sqrt{2}\int_0^t [\nabla^2\phi^*(Y_s)]^{-1/2}dW_s \tag{18}$$

given $x_k = \nabla\phi^*(Y_0)$ from the previous iteration. Here step (16) can again be solved iteratively using Euler-Maruyama and (17) is another convex optimization for which we can implement approximately.

We include the argument of this scheme in Appendix E, from which we see that one gets a better rate compared to the previous forward-discretized method ($\Omega(\delta^{-1/2})$ vs. $\Omega(\delta^{-1})$) while maintaining no bias. The crucial step, which was also used in the work of [23], is to relate the process (18) to an SDE for which the $\hat{\mu}$ and $G$ only involve samples at $X_t$ and no other time points (e.g., $X_0$), contrary to what happens for EM and forward discretization. This allows us to get a tighter control on the relevant quantities for bounding the discretization error and hence a better final rate (cf. Lemma 9). The difficulty for the other two schemes lies in the fact that due to the stochastic Brownian motion term, if we were to bound the discretization error between two time points $X_0$ and $X_t$, one would need a more global stability control (hence dependence on $M$) to account for the small probability that they are far apart (therefore local stability implied by self-concordance doesn't help). Our analysis for the algorithm works with the smoothness assumption stated below.

**Assumption 7** (Weaker $\gamma$-Relative Smooth). *For all $x, x' \in dom(\phi)$, it holds that*

$$-\gamma\nabla^2\phi(x) \preceq \nabla^2 f(x) \preceq \gamma\nabla^2\phi(x),$$

$$\max\left\{\|\nabla^2 f(x)[\nabla^2\phi(x)]^{-1} - \nabla^2 f(x')[\nabla^2\phi(x')]^{-1}\|_{op},\right.$$
$$\left.\|[\nabla^2\phi(x)]^{-1}\nabla^2 f(x) - [\nabla^2\phi(x')]^{-1}\nabla^2 f(x')\|_{op}\right\} \leq K\|x - x'\|.$$

Examples abound for such an assumption. We give an example here where $f$ is not smooth, yet satisfy this relative smoothness condition above. For $f(x) = x\log(x)$ which does not have Lipschitz gradient, picking the strongly convex $\phi(x) = x\log(x) + (1-x)\log(1-x)$ where $dom(\phi) = [0,1]$, it's easy to see $f''/\phi'' = x^{-1}/(x^{-1} + (1-x)^{-1})$ satisfy both requirements.

Below we give the result for our algorithm. As alluded to earlier, this new proposal removes the $M$ dependence altogether. It is somewhat expected that relative-smoothness type assumption would show up in the result, the lack of which necessarily implies that the potential $f$ is "misaligned" w.r.t the underlying changing geometry, for which sampling is unequivocally expected to be hard. But what's surprising is that this is in fact all that's required from $\phi$ for our method, and $\phi$ in itself doesn't have to be smooth or self-concordant in this case.

**Proposition 4** (Convergence Guarantee for Backward Discretization). *For the Algorithm (16)-(17), under Assumption 2,3,5,7 and stepsize $h = \mathcal{O}(\min\{1/\gamma, 1/K, 1/\beta, \sqrt{\delta\beta/(\gamma^2 L^2 + \alpha^{-1}d^3 K^2)}\})$, after $k \geq \tilde{\Omega}(\sqrt{\gamma^2 L^2 + \alpha^{-1}d^3 K^2}/\delta^{1/2}\beta^{3/2})$ iterations, we have $H_\pi(\rho_k) \leq \delta$.*

Let us mention in passing some possible extensions to the framework presented above. In the case when we are dealing with potential with finite sum structure, i.e., $f(x) = \sum_i f_i(x)$, as is often the case in machine learning problems, one could execute instead of (17) the update

$$x_{k+1} = \arg\min_v \eta\sum_{i\in B} f_i(v) + \phi(v) - y_\eta^\top v, \tag{19}$$

where $B$ is a random batch of data points. Such algorithm effectively assumes that we have stochastic (and therefore) noisy access to $\nabla f$, where $\hat{\nabla} f(x) = \nabla f(x) + \zeta$ for $\zeta$ an independent random noise vector with $\mathbb{E}[\zeta] = 0$ and $\mathbb{E}[\|\zeta\|_2^2] \le d\sigma^2$. Basic considerations suggest that this stochastic variant of the algorithm will converge to a noise ball with radius that scales with the variance of the noise $\sigma$, but is nevertheless more efficient from a computational perspective.

## 5 Numerical Experiments

In this section, we test out the induced bias and the benefit of using mirror maps in two experiments.

For the first experiment, we take the similar setup as in [6], where we pick $\phi = f$ (i.e., Newton) and consider uniform sampling from a 2D box $[-0.01, 0.01] \times [-1, 1]$. Four methods are compared. For Newton Langevin, we aim to target $\pi_\beta \propto \exp(-\beta \cdot \phi)$, taking $\phi(x) = -\log(1 - x_1^2) - \log(0.01^2 - x_2^2)$ as the barrier. We test out the 3 different discretization schemes with $\beta = 10^{-4}$ so that $\pi_\beta \approx \pi$. Stepsize is chosen to be $h = 10^{-5}$. Projected Langevin is taken to be another option for dealing with constraints, which targets the uniform distribution $\pi$ directly and simply performs ULA followed by projection onto the domain. The plot below shows the samples after 500 iterations, where $\nabla \phi^*$ and the proximal operator are solved with 50 steps of gradient descent steps. Diffusion term $\phi$ is solved with 10 inner steps of EM. From the samples, EM seems to give qualitatively different result, suggesting the possible existence of bias.

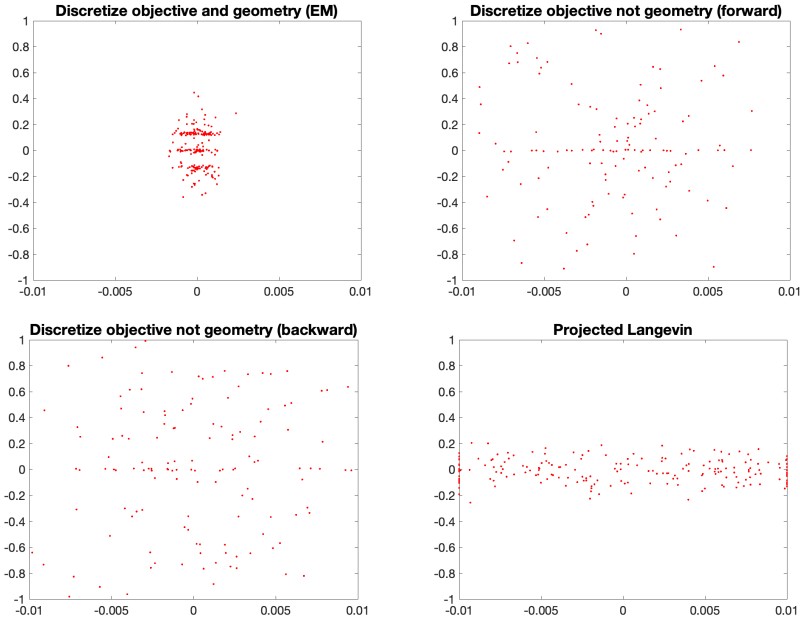

Figure 1: Uniform sampling from ill-conditioned box $[-0.01, 0.01] \times [-1, 1]$.

The second experiment concerns ill-conditioned Gaussian potential (no bias in this case), for which we compare the speed of convergence for Mirror-Langevin (with EM discretization) vs. ULA. We take $\phi = f = (x - \mu)^\top \Sigma^{-1} (x - \mu)/2$, and repeat the process 200 times with $d = 50$, computing the error in empirical mean $\|\hat{\mu} - \mu\|_2$ and covariance $\|\hat{\Sigma} - \Sigma\|_F / \|\Sigma\|_F$ and plot them across iterations below. Stepsize $h$ is picked to be $10^{-3}$ in both cases and initialization as $\mathcal{N}(0, I)$.

## 6 Discussion

Our result characterizes the interplay between $\phi$ and $f$ for different discretization schemes, which can be used to guide particular choice of mirror map given the sampling problem on hand. Our newly proposed algorithm and the analysis of several previously proposed schemes in the setting of

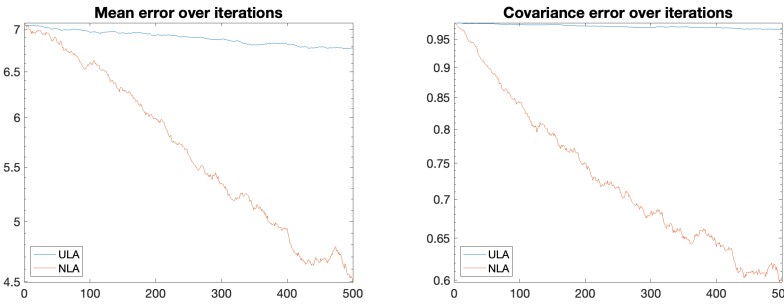

Figure 2: Newton Langevin vs. ULA for Gaussian ($\Sigma = \text{diag}(1, 2, \cdots, 50), \mu = [1, \cdots, 1]$).

sampling from nonsmooth and nonconvex potentials highlight several interesting distinctions that do not find parallel in the traditional Euclidean setup.

In optimization, one typically requires quite strong assumption for global stable convergence of Newton's method (for which $\phi = f$). Are various tricks such as Trust-Region, Cubic-Regularized Newton justified here as well for either speeding up sampling or correcting for bias? As future steps, it is an unfulfilled dream of ours to formalize and confirm a lower bound. It is conceivable that the extra bias term is unavoidable and captures the price we have to pay for discretizing $\phi$ while asking for weaker smoothness. It's also interesting to ask whether higher-order discretizers such as Runge-Kutta would yield better rate. In a different vein, one could explore the possibility of higher-order dynamics (à la underdamped Langevin) for which more sophisticated integrator [19, 16] could potentially be leveraged. Conventional wisdom suggests that the introduction of auxiliary variable that handles the non-smoothness of Brownian motion can often lead to design of better discrete sampling algorithms. On the probability theory front, it remains an intriguing open question to give a complete characterization for the Mirror-LSI condition.

## Acknowledgments and Disclosure of Funding

We are grateful for the constructive comments and feedback from the anonymous reviewers. This work was partially supported under NSF-2032014.

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
