# A Properties of Mirror Langevin SDE

## A.1 Equivalence between (2) and (3)

Let $T(Y) = \nabla\phi^*(Y)$, and using that (1) $X_t = \nabla\phi^*(Y_t)$; (2) $\nabla^2\phi^*(Y_t) = [\nabla^2\phi(X_t)]^{-1}$, Itô's Lemma using equation (2) gives

$$dX_t = dT(Y_t) = -\nabla T(Y_t)^\top \nabla f(X_t)dt + \mathrm{Tr}\left(\nabla^2\phi(X_t)\nabla^2 T(Y_t)\right)dt + \sqrt{2}\nabla T(Y_t)^\top\sqrt{\nabla^2\phi(X_t)}dW_t\,.$$

Moreover, we have

$$\nabla T(Y) = [\nabla^2\phi(X)]^{-1} \tag{20}$$

$$\nabla^2 T(Y) = -[\nabla^2\phi(X)]^{-1}\frac{d\nabla^2\phi(X)}{dY}[\nabla^2\phi(X)]^{-1}$$

Therefore

$$dX_t = -[\nabla^2\phi(X_t)]^{-1}\nabla f(X_t)dt - \mathrm{Tr}\left(\frac{d\nabla^2\phi(X_t)}{dY_t}[\nabla^2\phi(X_t)]^{-1}\right)dt + \sqrt{2[\nabla^2\phi(X_t)]^{-1}}dW_t\,.$$

For the Trace operation on tensor-matrix product, we define it as $\mathrm{Tr}(\nabla^3\phi(X)G(X)) \in \mathbb{R}^d$, where the $i$-th element is $\mathrm{Tr}(\nabla_i\nabla^2\phi(X)G(X)) = \sum_{j,k}\frac{\partial^3\phi(X)}{\partial X_i\partial X_j\partial X_k}G(X)_{j,k}$. Looking at the $i$-the coordinate, the middle trace term becomes

$$\mathrm{Tr}\left([\nabla^2\phi(X)]^{-1}\sum_j\frac{\partial\nabla^2\phi(X)}{\partial X(j)}\frac{\partial X(j)}{\partial Y(i)}\right)$$

$$= \sum_j\mathrm{Tr}\left([\nabla^2\phi(X)]^{-1}\nabla_j\nabla^2\phi(X)\right)[\nabla^2\phi(X)]^{-1}_{i,j}$$

where we used that $X = T(Y)$ and equation (20). This is, of course, equal to, looking again at $i$-th element,

$$e_i^\top[\nabla^2\phi(X)]^{-1}\mathrm{Tr}\left(\nabla^3\phi(X)[\nabla^2\phi(X)]^{-1}\right)$$

$$= \sum_j[\nabla^2\phi(X)]^{-1}_{i,j}\mathrm{Tr}\left(\nabla_j\nabla^2\phi(X)[\nabla^2\phi(X)]^{-1}\right)\,.$$

Therefore

$$dX_t = -[\nabla^2\phi(X_t)]^{-1}\nabla f(X_t)dt - [\nabla^2\phi(X_t)]^{-1}\mathrm{Tr}\left(\nabla^3\phi(X_t)[\nabla^2\phi(X_t)]^{-1}\right)dt + \sqrt{2[\nabla^2\phi(X_t)]^{-1}}dW_t\,,$$

as claimed in equation (3).

## A.2 Stationary distribution

Lemma 3 from [23] shows that for the SDE (under assumption $\nabla^2\phi \succ 0$)

$$dX_t = \left(\nabla\cdot[\nabla^2\phi(X_t)]^{-1} - [\nabla^2\phi(X_t)]^{-1}\nabla f(X_t)\right)dt + \sqrt{2[\nabla^2\phi(X_t)]^{-1}}dW_t\,, \tag{21}$$

the density $X_t \sim \rho_t$ satisfies the Fokker-Planck equation

$$\frac{\partial\rho_t}{\partial t} = \nabla\cdot\left(\rho_t[\nabla^2\phi]^{-1}\nabla\log\frac{\rho_t}{\pi}\right)\,, \tag{22}$$

from which it is evident that $\pi = e^{-f}$ is an invariant measure. Comparing (21) and (3), it suffices to show

$$\nabla\cdot[\nabla^2\phi(X_t)]^{-1} = -[\nabla^2\phi(X_t)]^{-1}\mathrm{Tr}\left(\nabla^3\phi(X_t)[\nabla^2\phi(X_t)]^{-1}\right)\,. \tag{23}$$

Looking at the $i$-th element, we have

$$\sum_j\frac{\partial[\nabla^2\phi(X)]^{-1}_{i,j}}{\partial X(j)} = -\sum_j\sum_{s,t}[\nabla^2\phi(X)]^{-1}_{i,s}[\nabla^2\phi(X)]^{-1}_{t,j}\nabla_s[\nabla^2\phi(X)]_{t,j}$$

$$= -\sum_s[\nabla^2\phi(X)]^{-1}_{i,s}\mathrm{Tr}\left(\nabla_s\nabla^2\phi(X)[\nabla^2\phi(X)]^{-1}\right)\,,$$

same as RHS. This also concludes that the density of $X_t$ from (3) follows the PDE (22).

# B Properties of Mirror LSI

**Lemma 2** (Stability under bounded perturbation for Mirror LSI). *Suppose probability density $\pi$ satisfies Mirror-LSI with parameter $\beta$, then provided $\epsilon \le \frac{d\nu}{d\pi} \le \delta$, for $\epsilon, \delta > 0$, the probability density $\nu$ satisfies Mirror-LSI with parameter $\delta/\beta\epsilon$.*

*Proof.* We denote the RHS of (5) as $\mathrm{Ent}_\pi[g^2]$. One could show using the variational principal of entropy that for $\nu \ll \pi$

$$\mathrm{Ent}_\nu[g^2] \le \left\| \frac{d\nu}{d\pi} \right\|_\infty \mathrm{Ent}_\pi[g^2] \le \delta \cdot \mathrm{Ent}_\pi[g^2].$$

where we used the assumption $\epsilon \le \frac{d\nu}{d\pi} \le \delta$, for $\epsilon, \delta > 0$. For the LHS,

$$\frac{2}{\beta} \int \|\nabla g(x)\|^2_{[\nabla^2 \phi(x)]^{-1}} d\pi = \frac{2}{\beta} \int \|\nabla g(x)\|^2_{[\nabla^2 \phi(x)]^{-1}} \frac{d\pi}{d\nu} d\nu$$

$$\le \frac{2}{\beta\epsilon} \int \|\nabla g(x)\|^2_{[\nabla^2 \phi(x)]^{-1}} d\nu$$

Putting things together, we have

$$\frac{2\delta}{\beta\epsilon} \int \|\nabla g(x)\|^2_{[\nabla^2 \phi(x)]^{-1}} d\nu \ge \mathrm{Ent}_\nu[g^2]$$

as claimed. □

There's an analogous and equivalent version of Mirror LSI (and Talagrand's inequality) in the dual $y$-space (as shown below) that can drive exponential convergence in the continuous dynamics (9), from which we can try to bound the discretization error. Similar analysis in the following sections, of course, can be carried out there which gives similar convergence result to $(\nabla\phi)_{\#}\pi$ in the dual space.

**Lemma 3** (Variant of Talagrand's inequality). *Mirror LSI for $\pi$ as in (6) implies a generalized Talagrand's inequality, i.e., for all $\rho$,*

$$\frac{\beta}{2} W_{2,[\nabla^2\phi]^{-1}}(\nabla\phi_{\#}\rho, \nabla\phi_{\#}\pi)^2 \le H_\pi(\rho),$$

*where $W_{2,[\nabla^2\phi]^{-1}}(\nabla\phi_{\#}\rho, \nabla\phi_{\#}\pi)^2 := \inf_{x\sim\rho, x'\sim\pi} \mathbb{E}[\|\nabla\phi(x) - \nabla\phi(x')\|^2_{[\nabla^2\phi]^{-1}}].$*

*Proof.* Mirror LSI with parameter $\beta$ for $\pi$ in $x$-space implies that the density $\nabla\phi_{\#}\pi$ satisfies a dual Mirror-LSI with parameter $\beta$ in $y$-space , i.e., for all $\rho$,

$$\int \nabla\phi_{\#}\rho(x) \log \frac{\nabla\phi_{\#}\rho(x)}{\nabla\phi_{\#}\pi(x)} dx \le \frac{1}{2\beta} \int \nabla\phi_{\#}\rho(x) \left\| \nabla \log \frac{\nabla\phi_{\#}\rho(x)}{\nabla\phi_{\#}\pi(x)} \right\|^2_{\nabla^2\phi} dx. \tag{24}$$

We begin by showing that the LHS is invariant under bijective mapping $\nabla\phi : \mathbb{R}^d \to \mathbb{R}^d$. Let $\rho' = \nabla\phi_{\#}\rho$ and $\pi' = \nabla\phi_{\#}\pi$, then the change of variable formula gives

$$\frac{\rho(x)}{\pi(x)} = \frac{\rho'(\nabla\phi(x)) \det(\nabla^2\phi(x))}{\pi'(\nabla\phi(x)) \det(\nabla^2\phi(x))} = \frac{\rho'(\nabla\phi(x))}{\pi'(\nabla\phi(x))}.$$

Therefore since $\nabla\phi(x) \sim \rho'$ if $x \sim \rho$, we have

$$\mathbb{E}_{x\sim\rho'} \left[ \log \frac{\rho'(x)}{\pi'(x)} \right] = \mathbb{E}_{x\sim\rho} \left[ \log \frac{\rho'(\nabla\phi(x))}{\pi'(\nabla\phi(x))} \right] = \mathbb{E}_{x\sim\rho} \left[ \log \frac{\rho(x)}{\pi(x)} \right].$$

The RHS follows by observing that if $x \sim \rho$, $y = \nabla\phi(x) \sim \rho'$. Let $h(x) = \log \frac{\rho(x)}{\pi(x)}$ and $\tilde{h}(y) = h(\nabla\phi^*(y))$,

$$\mathbb{E}_{x\sim\rho} \left[ \|\nabla_x h(x)\|^2_{[\nabla^2\phi(x)]^{-1}} \right] = \mathbb{E}_{y\sim\rho'} \left[ \left\| \nabla_{\nabla\phi^*(y)} \tilde{h}(y) \right\|^2_{[\nabla^2\phi(\nabla\phi^*(y))]^{-1}} \right]$$

$$= \mathbb{E}_{y \sim \rho'} \left[ \left\| \nabla^2 \phi(x) \nabla_y \tilde{h}(y) \right\|^2_{[\nabla^2 \phi(x)]^{-1}} \right]$$

$$= \mathbb{E}_{y \sim \rho'} \left[ \left\| \nabla_y \tilde{h}(y) \right\|^2_{\nabla^2 \phi(x)} \right]$$

where we used $\nabla_y \tilde{h}(y) = [\nabla_y \nabla \phi^*(y)]^\top \nabla_{\nabla \phi^*(y)} \tilde{h}(y) = [\nabla^2 \phi(x)]^{-1} \nabla_{\nabla \phi^*(y)} \tilde{h}(y)$. Having established (24), applying the manifold version of LSI $\Rightarrow$ Talagrand's inequality (cf. Theorem 22.17 in [21]) on $\nabla \phi_\# \pi$, we therefore have

$$\frac{\beta}{2} W_{2,[\nabla^2 \phi]^{-1}} (\nabla \phi_\# \rho, \nabla \phi_\# \pi)^2 \leq H_{\nabla \phi_\# \pi}(\nabla \phi_\# \rho) = H_\pi(\rho) .$$

$\square$

## C   Proofs for Section 4.2: EM Discretization

*Proof of Lemma 1.* Using (9) and let $X = T(Y) := \nabla \phi^*(Y)$, then Itô's Lemma gives

$$dX_t = dT(Y_t) = -\nabla T(Y_t)^\top \nabla f(X_0) dt + \text{Tr} \left( \nabla^2 \phi(X_0) \nabla^2 T(Y_t) \right) dt + \sqrt{2} \nabla T(Y_t)^\top \sqrt{\nabla^2 \phi(X_0)} dW_t .$$

From here, a similar calculation as in Appendix A.1 shows

$$dX_t = -[\nabla^2 \phi(X_t)]^{-1} \nabla f(X_0) dt - [\nabla^2 \phi(X_t)]^{-1} \text{Tr} \left( \nabla^3 \phi(X_t) [\nabla^2 \phi(X_t)]^{-1} \nabla^2 \phi(X_0) [\nabla^2 \phi(X_t)]^{-1} \right) dt$$
$$+ \sqrt{2} [\nabla^2 \phi(X_t)]^{-1} \sqrt{\nabla^2 \phi(X_0)} dW_t ,$$

where we can easily identify $G$ and

$$\hat{\mu} = -[\nabla^2 \phi(X_t)]^{-1} \nabla f(X_0) - [\nabla^2 \phi(X_t)]^{-1} \text{Tr} \left( \nabla^3 \phi(X_t) [\nabla^2 \phi(X_t)]^{-1} \nabla^2 \phi(X_0) [\nabla^2 \phi(X_t)]^{-1} \right)$$
$$- \nabla \cdot G + G \nabla f(X_t)$$

as claimed. $\square$

Below is a helper lemma stating the implication of self-concordance assumption on Hessian stability. Being an affine invariant property, it is a more natual assumption compared to those made in previous works, i.e., Lipschitz Hessian.

**Lemma 4** (Self-Concordance Implication)**.** *Under Assumption 1, 3 and 5, we have for D the diameter* $D := \max_{u,v} \|\nabla \phi(u) - \nabla \phi(v)\|_2$ *and updates* $X_t = \nabla \phi^*(Y_t)$ *and* $X_0 = \nabla \phi^*(Y_0)$ *following* (9),

$$M^{-1} \cdot [\nabla^2 \phi(X_t)]^{-1} \preceq [\nabla^2 \phi(X_t)]^{-1} \nabla^2 \phi(X_0) [\nabla^2 \phi(X_t)]^{-1} \preceq M \cdot [\nabla^2 \phi(X_t)]^{-1}$$

*in expectation (w.r.t Brownian motion) for*

$$M = (1 - \exp(-1/16\zeta^2 t)) \cdot (1 - \zeta(tL + 2\sqrt{td}))^{-2} + \exp(-1/16\zeta^2 t) \cdot \exp(2\zeta D/\sqrt{\alpha}) ,$$

*if* $t \leq \min(1/2\zeta L, 1/16\zeta^2 d)$ *and bounded by*

$$M = \exp(2\zeta D/\sqrt{\alpha})$$

*deterministically. We use the convention* $M = 1$ *when* $\zeta = 0$ *and* $D = \infty$. *Moreover, it implies*

$$\| \text{Tr} \left( \nabla^3 \phi(X_t) [\nabla^2 \phi(X_t)]^{-1} \right) \|^2_{[\nabla^2 \phi(X_t)]^{-1}} \leq 4\zeta^2 d^2 .$$

*Proof.* From the definition of self-concordance for $\phi^*$, it implies from [18] that for $\|Y_t - Y_0\|_{\nabla^2 \phi^*(Y_0)} \leq 1/\zeta$, we have

$$\left( 1 - \zeta \|Y_t - Y_0\|_{\nabla^2 \phi^*(Y_0)} \right)^2 \nabla^2 \phi^*(Y_0) \preceq \nabla^2 \phi^*(Y_t) \preceq \left( 1 - \zeta \|Y_t - Y_0\|_{\nabla^2 \phi^*(Y_0)} \right)^{-2} \nabla^2 \phi^*(Y_0) .$$

Therefore since $\nabla \phi(X_t) - \nabla \phi(X_0) = Y_t - Y_0 = -t \cdot \nabla f(X_0) + \sqrt{2t \nabla^2 \phi(X_0)} \cdot z_0$, and $z_0$ is independent of everything else,

$$\|Y_t - Y_0\|^2_{\nabla^2 \phi^*(Y_0)} = \left\| -t \cdot \nabla f(X_0) + \sqrt{2t \nabla^2 \phi(X_0)} \cdot z_0 \right\|^2_{[\nabla^2 \phi(X_0)]^{-1}}$$

$$= t^2 \|\nabla f(X_0)\|^2_{[\nabla^2\phi(X_0)]^{-1}} + 2t\|z_0\|^2_2$$
$$\leq t^2 L^2 + 2t\|z_0\|^2_2$$

Using $\chi^2$ concentration, $\mathbb{P}(\|z_0\|^2_2 \geq (\sqrt{d} + \sqrt{\delta})^2) \leq \exp(-\delta)$, for $t \leq \min(1/2\zeta L, 1/16\zeta^2 d)$, with probability at least $1 - \exp(-d) \geq 1 - \exp(-1/16\zeta^2 t)$ over the draw of $z_0$, we have $\|Y_t - Y_0\|_{\nabla^2\phi^*(Y_0)} \leq tL + 2\sqrt{td} < 1/\zeta$, therefore

$$(1 - \zeta(tL + 2\sqrt{td}))^2 \cdot I \preceq \nabla^2\phi(X_0)^{1/2}[\nabla^2\phi(X_t)]^{-1}\nabla^2\phi(X_0)^{1/2} \preceq (1 - \zeta(tL + 2\sqrt{td}))^{-2} \cdot I.$$

One the other hand, with the remaining probability $\exp(-1/16\zeta^2 t)$, consider the function $g(s) = u^\top \nabla^2\phi^*(Y_0 + s(Y_t - Y_0))u =: \nabla^2\phi^*(Y_s)[u, u]$, then from self concordance we have

$$|g'(s)| = |\nabla^3\phi^*(Y_s)[u, u, Y_t - Y_0]| \leq 2\zeta\|Y_t - Y_0\|_{\nabla^2\phi^*(Y_s)}\|u\|^2_{\nabla^2\phi^*(Y_s)}$$
$$= 2\zeta\|Y_t - Y_0\|_{\nabla^2\phi^*(Y_s)}g(s)$$
$$\leq \frac{2\zeta}{\sqrt{\alpha}}\|Y_t - Y_0\|_2 \cdot g(s)$$
$$\leq \frac{2\zeta}{\sqrt{\alpha}}D \cdot g(s)$$

therefore $|\log(g(1)) - \log(g(0))| \leq 2\zeta D/\sqrt{\alpha}$ implying $\exp(-2\zeta D/\sqrt{\alpha})\nabla^2\phi^*(Y_0) \preceq \nabla^2\phi^*(Y_t) \preceq \exp(2\zeta D/\sqrt{\alpha})\nabla^2\phi^*(Y_0)$ and

$$\exp(-2\zeta D/\sqrt{\alpha}) \cdot I \preceq \nabla^2\phi(X_0)^{1/2}[\nabla^2\phi(X_t)]^{-1}\nabla^2\phi(X_0)^{1/2} \preceq \exp(2\zeta D/\sqrt{\alpha}) \cdot I.$$

Altogether this gives that in expectation w.r.t $z_0$, the stability parameter $M$ is upper bounded by

$$(1 - \exp(-1/16\zeta^2 t)) \cdot (1 - \zeta(tL + 2\sqrt{td}))^{-2} + \exp(-1/16\zeta^2 t) \cdot \exp(2\zeta D/\sqrt{\alpha})$$

which goes to 1 as $t \to 0$. Self concordance also implies picking direction $[[\nabla^2\phi(X_t)]^{1/2}e_i, u, u]$

$$-2\zeta\|e_i\|_2\nabla^2\phi^*(Y_t) \preceq \nabla^3\phi^*(Y_t)[[\nabla^2\phi(X_t)]^{1/2}e_i] \preceq 2\zeta\|e_i\|_2\nabla^2\phi^*(Y_t)$$

which means using the derivation in A.1 that $\left\|\sum_j [\nabla^2\phi(X_t)]^{-1/2}_{ij}[\nabla^2\phi(X_t)]^{-1}\nabla_j\nabla^2\phi(X_t)\right\|_{op} \leq 2\zeta \ \forall i \in [d]$, therefore

$$|\sum_j [\nabla^2\phi(X_t)]^{-1/2}_{ij} \operatorname{Tr}([\nabla^2\phi(X_t)]^{-1}\nabla_j\nabla^2\phi(X_t))| \leq 2\zeta\sqrt{d}$$

and we have

$$\|\operatorname{Tr}\left(\nabla^3\phi(X_t)[\nabla^2\phi(X_t)]^{-1}\right)\|^2_{[\nabla^2\phi(X_t)]^{-1}}$$
$$= \|[\nabla^2\phi(X_t)]^{-1/2}\operatorname{Tr}\left(\nabla^3\phi(X_t)[\nabla^2\phi(X_t)]^{-1}\right)\|^2_2$$
$$\leq d|\sum_j [\nabla^2\phi(X_t)]^{-1/2}_{ij}\operatorname{Tr}\left(\nabla_j\nabla^2\phi(X_t)[\nabla^2\phi(X_t)]^{-1}\right)|^2$$
$$\leq 4\zeta^2 d^2$$

as desired. $\qquad\square$

We collect some useful results before giving the per-iteration progress bound.

**Lemma 5** (Control on $\|\hat{\mu}\|^2_{\nabla^2\phi}$). *Under Assumption 1 and 4, we have for the $\hat{\mu}$ defined in Lemma 1*

$$\|\hat{\mu}\|^2_{\nabla^2\phi} \leq 2\eta\gamma^2\|\nabla\phi(X_t) - \nabla\phi(X_0)\|^2_{[\nabla^2\phi(X_0)]^{-1}} + 8\eta^2\zeta^2 d^2,$$

*where we denote $\eta = \|[\nabla^2\phi(X_t)]^{-1}[\nabla^2\phi(X_0)]\|_{op}$.*

*Proof.* Let $v := \nabla \cdot \left([\nabla^2\phi(X_t)]^{-1}\nabla^2\phi(X_0)[\nabla^2\phi(X_t)]^{-1}\right)$, using the fact that

$$\frac{\partial \operatorname{Tr}((X^\top CX)^{-1}A)}{\partial X} = -(CX(X^\top CX)^{-1})(A + A^\top)(X^\top CX)^{-1},$$

the $i$-th element of $v$ is

$$-\sum_j \sum_{s,t} \left( [\nabla^2\phi(X_t)]_{s,j}^{-1} \left( [\nabla^2\phi(X_t)]^{-1}\nabla^2\phi(X_0)[\nabla^2\phi(X_t)]^{-1} \right)_{i,t} \right.$$

$$\left. + [\nabla^2\phi(X_t)]_{s,i}^{-1} \left( [\nabla^2\phi(X_t)]^{-1}\nabla^2\phi(X_0)[\nabla^2\phi(X_t)]^{-1} \right)_{j,t} \right) \frac{\partial \nabla^2(X_t)_{s,t}}{\partial X_t(j)}$$

$$= -\sum_s \left( [\nabla^2\phi(X_t)]^{-1} \right)_{i,s} \operatorname{Tr}\left( \nabla_s\nabla^2\phi(X_t)[\nabla^2\phi(X_t)]^{-1}\nabla^2\phi(X_0)\nabla^2\phi(X_t)]^{-1} \right)$$

$$- \sum_t \left( [\nabla^2\phi(X_t)]^{-1}\nabla^2\phi(X_0)[\nabla^2\phi(X_t)]^{-1} \right)_{i,t} \operatorname{Tr}\left( \nabla_t\nabla^2\phi(X_t)[\nabla^2\phi(X_t)]^{-1} \right) .$$

Therefore

$$v = -[\nabla^2\phi(X_t)]^{-1}\operatorname{Tr}\left( \nabla^3\phi(X_t)[\nabla^2\phi(X_t)]^{-1}\nabla^2\phi(X_0)[\nabla^2\phi(X_t)]^{-1} \right)$$

$$- [\nabla^2\phi(X_t)]^{-1}\nabla^2\phi(X_0)[\nabla^2\phi(X_t)]^{-1}\operatorname{Tr}\left( \nabla^3\phi(X_t)[\nabla^2\phi(X_t)]^{-1} \right) .$$

Putting together with the expression in Lemma 1, $\|\hat\mu\|_{\nabla^2\phi}^2$ is

$$\left\| \nabla^2\phi(X_0)[\nabla^2\phi(X_t)]^{-1}\nabla f(X_t) - \nabla f(X_0) + \nabla^2\phi(X_0)[\nabla^2\phi(X_t)]^{-1}\operatorname{Tr}\left( \nabla^3\phi(X_t)[\nabla^2\phi(X_t)]^{-1} \right) \right\|_{[\nabla^2\phi(X_t)]^{-1}}^2$$

$$\leq 2\left\| [\nabla^2\phi(X_t)]^{-1/2}[\nabla^2\phi(X_0)]^{1/2} \right\|_{op}^2 \left\| [\nabla^2\phi(X_0)]^{1/2}[\nabla^2\phi(X_t)]^{-1}\nabla f(X_t) - [\nabla^2\phi(X_0)]^{-1/2}\nabla f(X_0) \right\|_2^2$$

$$+ 2\left\| [\nabla^2\phi(X_t)]^{-1/2}\nabla^2\phi(X_0)[\nabla^2\phi(X_t)]^{-1}\operatorname{Tr}\left( \nabla^3\phi(X_t)[\nabla^2\phi(X_t)]^{-1} \right) \right\|_2^2$$

$$\leq 2\eta\gamma^2\|\nabla\phi(X_t) - \nabla\phi(X_0)\|_{[\nabla^2\phi(X_0)]^{-1}}^2 + 2\eta^2\|[\nabla^2\phi(X_t)]^{-1/2}\operatorname{Tr}\left( \nabla^3\phi(X_t)[\nabla^2\phi(X_t)]^{-1} \right)\|_2^2$$

$$\leq 2\eta\gamma^2\|\nabla\phi(X_t) - \nabla\phi(X_0)\|_{[\nabla^2\phi(X_0)]^{-1}}^2 + 8\eta^2\zeta^2d^2$$

where we used relative smoothness assumption 4 and Lemma 4. The first term will go to zero as $t \to 0$, whereas the second term will be responsible for the non-vanishing bias w.r.t the diminishing step size (as long as $\nabla^3\phi \neq 0$ so $\zeta \neq 0$). $\qquad\square$

Now we are ready to state the main recursion, drawing doses of inspiration from [23].

*Proof of Proposition 2.* Denoting $G_0(x)$ and $\hat\mu_0(x)$ as the diffusion/drift term at time $t$ when $x_t = x$ with $x_0$ at time $t = 0$, the Fokker-Planck equation for the conditional density $\rho_{t|0}(x_t|x_0)$ takes the form written below

$$\frac{\partial\rho_t(x)}{\partial t} = \int \frac{\partial\rho_{t|0}(x|x_0)}{\partial t}\rho_0(x_0)dx_0$$

$$= \int \left[ -\nabla\cdot(\rho_{t|0}(\nabla\cdot G_0(x) - G_0(x)\nabla f(x))) + \langle\nabla^2, \rho_{t|0}G_0(x)\rangle - \nabla\cdot(\rho_{t|0}\hat\mu_0(x)) \right] \rho_0(x_0)dx_0$$

$$= \nabla\cdot\left( \rho_{0|t}\int -(\rho_t(\nabla\cdot G_0(x) - G_0(x)\nabla f(x))) + \nabla\cdot(\rho_t G_0(x))dx_0 \right) - \nabla\cdot\left( \rho_t\int \rho_{0|t}\hat\mu_0(x)dx_0 \right)$$

$$= \nabla\cdot\left( \rho_{0|t}\int G_0(x)\nabla\rho_t + \rho_t G_0(x)\nabla f(x)dx_0 \right) - \nabla\cdot\left( \rho_t\int \rho_{0|t}\hat\mu_0(x)dx_0 \right)$$

$$= \nabla\cdot\left( \rho_{0|t}\int \left( \rho_t G_0(x)\nabla\log\frac{\rho_t}{\pi(x)} \right)dx_0 \right) - \nabla\cdot\left( \rho_t\underbrace{\int \rho_{0|t}\hat\mu_0(x)dx_0}_{\mathbb{E}_{\rho_{0|t}}[\hat\mu(x_0,x)|x_t=x],\text{func of } x} \right)$$

where for the second equality above we used Lemma 3 from [23] and (10). We will see that the first part corresponds to exponential decay to an unbiased limit (similar to what happens in Lemma 1) and the second corresponds to the biased shifted drift introduced by discretization. Let $M = \exp(2\zeta D/\sqrt\alpha)$, since $a^\top b = 2(\sqrt{M}a)^\top(\frac{1}{2\sqrt{M}}b) \leq M\|a\|_2^2 + \frac{1}{4M}\|b\|_2^2$ by Young's inequality,

$$\frac{d}{dt}H_\pi(\rho_t) = \int \frac{d\rho_t}{dt}\log\frac{\rho_t}{\pi}dx$$

$$= \int \nabla \cdot \left( \rho_{0|t} \int \rho_t G_0 \nabla \log \frac{\rho_t}{\pi(x)} dx_0 \right) \log \frac{\rho_t}{\pi} dx - \int \nabla \cdot \left( \rho_t \int \rho_{0|t} \hat{\mu}_0 dx_0 \right) \log \frac{\rho_t}{\pi} dx$$

$$= -\int \rho_{0|t} \int \rho_t \left\langle \nabla \log \frac{\rho_t}{\pi} G_0, \nabla \log \frac{\rho_t}{\pi} \right\rangle dx_0 dx + \int \rho_t \int \rho_{0|t} \langle \hat{\mu}_0, \nabla \log \frac{\rho_t}{\pi} \rangle dx_0 dx$$

$$= -\mathbb{E}_{\rho_{0,t}} \left[ \left\| \nabla \log \frac{\rho_t}{\pi} \right\|_G^2 \right] + \mathbb{E}_{\rho_{0,t}} \left[ \langle \hat{\mu}, \nabla \log \frac{\rho_t}{\pi} \rangle \right]$$

$$\leq -\frac{1}{M} \mathbb{E}_{\rho_t} \left[ \left\| \nabla \log \frac{\rho_t}{\pi} \right\|_{[\nabla^2 \phi]^{-1}}^2 \right] + \mathbb{E}_{\rho_{0,t}} \left[ \langle \hat{\mu}, \nabla \log \frac{\rho_t}{\pi} \rangle \right]$$

$$\leq -\frac{2\beta}{M} H_\pi(\rho_t) + M \mathbb{E}_{\rho_{0,t}} [\|\hat{\mu}\|_{\nabla^2 \phi}^2] + \frac{1}{4M} \mathbb{E}_{\rho_t} \left[ \left\| \nabla \log \frac{\rho_t}{\pi} \right\|_{[\nabla^2 \phi]^{-1}}^2 \right]$$

$$\leq -\frac{3\beta}{2M} H_\pi(\rho_t) + M \mathbb{E}_{\rho_{0,t}} [\|\hat{\mu}\|_{\nabla^2 \phi}^2]$$

where we used integration by parts, Mirror LSI and Lemma 4. Now using Lemma 5,

$$\mathbb{E}_{\rho_{0,t}} [\|\hat{\mu}\|_{\nabla^2 \phi}^2] \leq 2M\gamma^2 \mathbb{E}_{\rho_{0,t}} [\|\nabla \phi(x_t) - \nabla \phi(x_0)\|_{[\nabla^2 \phi(x_0)]^{-1}}^2] + 8\zeta^2 d^2 \mathbb{E}[\|\nabla^2 \phi(x_0)[\nabla^2 \phi(x_t)]^{-1}\|_{op}^2].$$

The first term can be bounded as (since $z_0$ is independent from $x_0$)

$$\mathbb{E}_{\rho_{0,t}} [\|y_t - y_0\|_{\nabla^2 \phi^*(y_0)}^2] = \mathbb{E}_{\rho_{0,t}} \left[ \left\| -t \cdot \nabla f(x_0) + \sqrt{2t \nabla^2 \phi(x_0)} \cdot z_0 \right\|_{[\nabla^2 \phi(x_0)]^{-1}}^2 \right]$$

$$= t^2 \mathbb{E}_{\rho_0} [\|\nabla f(x_0)\|_{[\nabla^2 \phi(x_0)]^{-1}}^2] + 2t \mathbb{E}[\|z_0\|_2^2]$$

$$\leq t^2 L^2 + 2td$$

and the second term is bounded using Lemma 4 as $8\zeta^2 d^2 \eta_t^2 := 8\zeta^2 d^2 ((1 - \exp(-1/16\zeta^2 t)) \cdot (1 - \zeta(tL + 2\sqrt{td}))^{-4} + \exp(-1/16\zeta^2 t) \cdot \exp(4\zeta D/\sqrt{\alpha}))$. Putting things together, we have for $\rho_t$ evolving according to (10), if $0 \leq t \leq h \leq \min(1/2\zeta L, 1/16\zeta^2 d, D/\sqrt{\alpha} L, D^2/4\alpha d)$,

$$\frac{d}{dt} H_\pi(\rho_t) \leq -\frac{3\beta}{2M} H_\pi(\rho_t) + 2M^2 \gamma^2 (t^2 L^2 + 2td) + 8M\zeta^2 d^2 \eta_t^2$$

$$\leq -\frac{3\beta}{2M} H_\pi(\rho_t) + 12M^2 \gamma^2 dh + 8M\zeta^2 d^2 \eta_h^2$$

where $\eta_h^2 \to 1$ as $h \to 0$. This can be rewritten as

$$\frac{d}{dt} \left( e^{\frac{3\beta}{2M} t} H_\pi(\rho_t) \right) \leq e^{\frac{3\beta}{2M} t} \left( 12M^2 \gamma^2 dh + 8M\zeta^2 d^2 \eta_h^2 \right).$$

Integrate it for $0 \leq t \leq h$, we have for $h \leq \frac{2M}{3\beta}$,

$$e^{\frac{3\beta}{2M} h} H_\pi(\rho_h) - H_\pi(\rho_0) \leq \frac{2M}{3\beta} (e^{\frac{3\beta h}{2M}} - 1)(12M^2 \gamma^2 dh + 8M\zeta^2 d^2 \eta_h^2)$$

$$\leq 24M^2 \gamma^2 dh^2 + 16M\zeta^2 d^2 \eta_h^2 h,$$

where we used $e^a \leq 1 + 2a$ for $a \in (0, 1]$. Therefore we end up with

$$H_\pi(\rho_h) \leq e^{-\frac{3\beta}{2M} h} H_\pi(\rho_0) + e^{-\frac{3\beta}{2M} h} (24M^2 \gamma^2 dh^2 + 16M\zeta^2 d^2 \eta_h^2 h)$$

$$\leq e^{-\frac{3\beta}{2M} h} H_\pi(\rho_0) + 24M^2 \gamma^2 dh^2 + 16M\zeta^2 d^2 \eta_h^2 h$$

and identifying $x_{k+1} \sim \rho_h$ and $x_k \sim \rho_0$ finishes the proof. $\qquad \square$

We have the following initial bound on the KL divergence with $x \sim \rho_0 = \mathcal{N}(x^*, \frac{1}{\gamma} I)$.

**Lemma 6** (Initialization). *Under Assumption 4 or 6 and Assumption 5, we have $H_\pi(\rho_0) \leq f(x^*) + \frac{d}{2} \log(\frac{\gamma}{2\pi e}) + \frac{\gamma}{2\alpha} \mathbb{E}_{\rho_0} [\|\nabla \phi(x^*) - \nabla \phi(x)\|_2^2]$, where $x^*$ satisfies $\nabla f(x^*) = 0$. Moreover, under Assumption 7, we have $H_\pi(\rho_0) \leq \frac{d}{2} \log(\frac{\gamma}{2\pi e}) + f(x^*) + \gamma \mathbb{E}_{\rho_0} [D_\phi(x, x^*)]$.*

*Proof.* From relative smoothness assumption 4 or 6, let $y_t = y + t(y^* - y)$ for $y, y^*$, where $x^* = \nabla\phi^*(y^*)$ is the staionary point (i.e., $\nabla f(x^*) = 0$). For $\tilde{f}(y) = f(\nabla\phi^*(y))$, this also gives $\nabla\tilde{f}(y^*) = 0$ and

$$
|\tilde{f}(y) - [\tilde{f}(y^*) + \langle\nabla\tilde{f}(y^*), y - y^*\rangle]|
$$
$$
= |\int_0^1 [\nabla\tilde{f}(y^*) - \nabla\tilde{f}(y_t)]^\top (y^* - y)\, dt|
$$
$$
\leq \int_0^1 \|[\nabla^2\phi(x^*)]^{-1}\nabla f(x^*) - [\nabla^2\phi(x_t)]^{-1}\nabla f(x_t)\|_{\nabla^2\phi(x_t)} \cdot \|y^* - y\|_{[\nabla^2\phi(x_t)]^{-1}}\, dt
$$
$$
\leq \gamma \int_0^1 \|y^* - y_t\|_{[\nabla^2\phi(x_t)]^{-1}} \cdot \|y^* - y\|_{[\nabla^2\phi(x_t)]^{-1}}\, dt
$$
$$
\leq \frac{\gamma}{2\alpha}\|y^* - y\|_2^2
$$

where we used Cauchy-Schwarz and $\phi$ being $\alpha$-strongly convex implies $\phi^*$ is $1/\alpha$-smooth. Therefore

$$
\tilde{f}(y) \leq \tilde{f}(y^*) + \langle\nabla\tilde{f}(y^*), y - y^*\rangle + \frac{\gamma}{2\alpha}\|y^* - y\|_2^2
$$

and rewriting,

$$
f(x) \leq f(x^*) + \frac{\gamma}{2\alpha}\|\nabla\phi(x^*) - \nabla\phi(x)\|_2^2. \tag{25}
$$

For $x \sim \rho_0 = \mathcal{N}(x^*, \frac{1}{\gamma}I)$ Gaussian centered at $x^*$, we have

$$
H_\pi(\rho_0) = -H(\rho_0) + \mathbb{E}_{\rho_0}[f] \leq \frac{d}{2}\log(\frac{\gamma}{2\pi e}) + f(x^*) + \frac{\gamma}{2\alpha}\mathbb{E}_{\rho_0}[\|\nabla\phi(x^*) - \nabla\phi(x)\|_2^2],
$$

where we used that for normal distribution $H(\rho_0) = \frac{d}{2}\log\frac{2\pi e}{\gamma}$.

With Assumption 7, instead of (25), we have

$$
f(x) = f(x^*) + \int_0^1 \int_0^t (x - x^*)^\top \nabla^2 f(x_s)(x - x^*) ds\, dt
$$
$$
\leq f(x^*) + \gamma \int_0^1 \int_0^t (x - x^*)^\top \nabla^2\phi(x_s)(x - x^*) ds\, dt
$$
$$
= f(x^*) + \gamma \int_0^1 (x - x^*)^\top (\nabla\phi(x_t) - \nabla\phi(x^*)) dt
$$
$$
= f(x^*) + \gamma [\phi(x) - \phi(x^*) - (x - x^*)^\top \nabla\phi(x^*)] =: f(x^*) + \gamma D_\phi(x, x^*)
$$

which gives

$$
H_\pi(\rho_0) = -H(\rho_0) + \mathbb{E}_{\rho_0}[f] \leq \frac{d}{2}\log(\frac{\gamma}{2\pi e}) + f(x^*) + \gamma\mathbb{E}_{\rho_0}[D_\phi(x, x^*)].
$$

$\square$

We end this section with a word about the relative smoothness assumptions.

**Lemma 7** (Relative Smoothness). *For the following conditions:*

1. $\nabla^2\tilde{f}(y) \preceq \gamma\nabla^2\phi^*(y)$

2. $\|[\nabla^2\phi(x)]^{-1}\nabla f(x) - [\nabla^2\phi(x')]^{-1}\nabla f(x')\|_{\nabla^2\phi(x')} \leq \gamma\|\nabla\phi(x) - \nabla\phi(x')\|_{[\nabla^2\phi(x')]^{-1}}$

3. $|\tilde{f}(y) - [\tilde{f}(y') + \langle\nabla\tilde{f}(y'), y - y'\rangle]| \leq \frac{\gamma}{2}\|y - y'\|_{\nabla^2\phi^*(y')}^2$

4. $\|\nabla f(x) - \nabla f(x')\|_{[\nabla^2\phi(x')]^{-1}} \leq \gamma\|\nabla\phi(x) - \nabla\phi(x')\|_{[\nabla^2\phi(x')]^{-1}}$

5. $-\gamma\nabla^2\phi(x) \preceq \nabla^2 f(x) \preceq \gamma\nabla^2\phi(x)$

where $y = \nabla\phi(x)$ and $\tilde{f}(y) = f(\nabla\phi^*(y))$, we have $2 \Rightarrow 1, 3$ and $4 \Rightarrow 5$. Moreover, taking $x = x^*$ for which $\nabla f(x^*) = 0$, condition 2 becomes the same as condition 4.

*Proof.* We begin by showing $\nabla^2 \tilde{f}(y) \succeq \gamma \nabla^2 \phi^*(y)$ implies 2 does not hold. Let $g(t) = \tilde{f}(y_t)$ for $y_t = y_0 + t \cdot z$, then for any $\epsilon$ there exists some $\delta$ such that $|1/\delta \cdot (g'(\delta) - g'(0)) - g''(0)| = |1/\delta \cdot (\nabla\tilde{f}(y_\delta) - \nabla\tilde{f}(y_0))^\top z - z^\top \nabla^2\tilde{f}(y_0)z| \leq \epsilon$, therefore we have from assumption

$$(\nabla\tilde{f}(y_\delta) - \nabla\tilde{f}(y_0))^\top z \geq \delta z^\top \nabla^2\tilde{f}(y_0)z - \epsilon\delta \geq \delta\gamma\|z\|^2_{\nabla^2\phi^*(y_0)} - \epsilon\delta\,,$$

from Cauchy-Schwarz for all $\epsilon'$

$$\|\nabla\tilde{f}(y_\delta) - \nabla\tilde{f}(y_0)\|_{[\nabla^2\phi^*(y_0)]^{-1}} \geq \delta\gamma\|z\|_{\nabla^2\phi^*(y_0)} - \epsilon' = \gamma\|y_\delta - y_0\|_{\nabla^2\phi^*(y_0)} - \epsilon'\,,$$

and chain rule $\nabla\tilde{f}(y) = [\nabla^2\phi(x)]^{-1}\nabla f(\nabla\phi^*(y))$ gives

$$\|[\nabla^2\phi(x_\delta)]^{-1}\nabla f(x_\delta) - [\nabla^2\phi(x_0)]^{-1}\nabla f(x_0)\|_{\nabla^2\phi(x_0)} \geq \gamma \cdot \|\nabla\phi(x_\delta) - \nabla\phi(x_0)\|_{[\nabla^2\phi(x_0)]^{-1}} - \epsilon'$$

for all $\epsilon'$, finishing the proof. This lets us conclude that $2 \Rightarrow 1$.

The proof of Lemma 6 shows that $2 \Rightarrow 3$.

For $4 \Rightarrow 5$, $\forall v \in \mathbb{R}^d$ we have

$$\begin{aligned}
\|\nabla^2 f(x)^\top v\|_{[\nabla^2\phi(x)]^{-1}} &= \lim_{h\to 0} \frac{\|\nabla f(x + vh) - \nabla f(x)\|_{[\nabla^2\phi(x)]^{-1}}}{h} \\
&\leq \lim_{h\to 0} \frac{\gamma \cdot \|\nabla\phi(x + vh) - \nabla\phi(x)\|_{[\nabla^2\phi(x)]^{-1}}}{h} \\
&= \gamma \cdot \|\nabla^2\phi(x)^\top v\|_{[\nabla^2\phi(x)]^{-1}}\,,
\end{aligned}$$

therefore $\nabla^2 f(x)^\top [\nabla^2\phi(x)]^{-1}\nabla^2 f(x) \preceq \gamma^2\nabla^2\phi(x)$, which in turn implies 5. $\qquad\square$

# D   Proofs for Section 4.3: Alternative Forward Discretization Scheme

*Proof of Proposition 3.* Following the derivation for Lemma 1, we have that (15) is the same as (in the primal space)

$$dX_t = -[\nabla^2\phi(X_t)]^{-1}\nabla f(X_0)dt - [\nabla^2\phi(X_t)]^{-1}\operatorname{Tr}\left(\nabla^3\phi(X_t)[\nabla^2\phi(X_t)]^{-1}\right)dt + \sqrt{2[\nabla^2\phi(X_t)]^{-1}}dW_t\,.$$

This means the process follows a weighted Langevin dynamics (10) with $G = [\nabla^2\phi(X_t)]^{-1}$ and $\hat{\mu} = [\nabla^2\phi(X_t)]^{-1}(\nabla f(X_t) - \nabla f(X_0))$ for (15) since (23) already taught us $\nabla \cdot G = -[\nabla^2\phi(X_t)]^{-1}\operatorname{Tr}\left(\nabla^3\phi(X_t)[\nabla^2\phi(X_t)]^{-1}\right)$.

Now tracing the proof of Proposition 2, we have from Mirror LSI

$$\begin{aligned}
\frac{d}{dt}H_\pi(\rho_t) &= \int \frac{d\rho_t}{dt}\log\frac{\rho_t}{\pi}dx \\
&= -\mathbb{E}_{\rho_t}\left[\left\|\nabla\log\frac{\rho_t}{\pi}\right\|^2_G\right] + \mathbb{E}_{\rho_{0,t}}\left[\langle\hat{\mu}, \nabla\log\frac{\rho_t}{\pi}\rangle\right] \\
&\leq -\mathbb{E}_{\rho_t}\left[\left\|\nabla\log\frac{\rho_t}{\pi}\right\|^2_{[\nabla^2\phi]^{-1}}\right] + \mathbb{E}_{\rho_{0,t}}[\|\hat{\mu}\|^2_{\nabla^2\phi}] + \frac{1}{4}\mathbb{E}_{\rho_t}\left[\left\|\nabla\log\frac{\rho_t}{\pi}\right\|^2_{[\nabla^2\phi]^{-1}}\right] \\
&\leq -\frac{3\beta}{2}H_\pi(\rho_t) + \mathbb{E}_{\rho_{0,t}}[\|\hat{\mu}\|^2_{\nabla^2\phi}]\,.
\end{aligned}$$

Using Assumption 1, 3, 5, 6 and (14), we have for $M = \exp(2\zeta D/\sqrt{\alpha})$, $\eta_t = (1 - \exp(-1/16\zeta^2 t)) \cdot (1 - \zeta(tL + 2\sqrt{td}))^{-2} + \exp(-1/16\zeta^2 t) \cdot M$,

$$\begin{aligned}
\mathbb{E}_{\rho_{0,t}}[\|\hat{\mu}\|^2_{\nabla^2\phi}] &\leq \gamma^2 \cdot \mathbb{E}_{\rho_{0,t}}\left[\|\nabla\phi(x_t) - \nabla\phi(x_0)\|^2_{[\nabla^2\phi(x_t)]^{-1}}\right] \\
&\leq \gamma^2 \cdot \mathbb{E}\left[\left\|-t\nabla f(x_0) + \sqrt{2}\int_0^t [\nabla^2\phi(x_s)]^{1/2}dW_s\right\|^2_{[\nabla^2\phi(x_t)]^{-1}}\right]
\end{aligned}$$

$$\leq 2\gamma^2 t^2 \eta_t L^2 + 4t\gamma^2 Md\,,$$

where we used Itô's isometry, $(a+b)^2 \leq 2a^2 + 2b^2$ and Lemma 4.

If $0 \leq t \leq h \leq \min(1/2\zeta L, 1/16\zeta^2 d, D/\sqrt{\alpha}L, D^2/4\alpha d)$,

$$\frac{d}{dt}H_\pi(\rho_t) \leq -\frac{3\beta}{2}H_\pi(\rho_t) + 2\gamma^2 h^2 \eta_h L^2 + 4h\gamma^2 Md\,.$$

Written differently,

$$\frac{d}{dt}\left(e^{\frac{3\beta}{2}t}H_\pi(\rho_t)\right) \leq e^{\frac{3\beta}{2}t}\left(2\gamma^2 h^2 \eta_h L^2 + 4h\gamma^2 Md\right)\,.$$

Integrate it for $0 \leq t \leq h$, we have for $h \leq \frac{1}{6\beta}$,

$$e^{\frac{3\beta}{2}h}H_\pi(\rho_h) - H_\pi(\rho_0) \leq \frac{2}{3\beta}(e^{\frac{3\beta h}{2}} - 1)(2\gamma^2 h^2 \eta_h L^2 + 4h\gamma^2 Md)$$
$$\leq 24\gamma^2 h^3 \eta_h L^2 + 16h^2\gamma^2 Md$$

where we used $e^a \leq 1 + 2a$ for $a \in (0, 1]$. Altogether this gives us

$$H_\pi(\rho_h) \leq e^{-\frac{3\beta}{2}h}H_\pi(\rho_0) + e^{-\frac{3\beta}{2}h}(24\gamma^2 h^3 \eta_h L^2 + 16h^2\gamma^2 Md)$$
$$\leq e^{-\frac{3\beta}{2}h}H_\pi(\rho_0) + 24\gamma^2 h^3 \eta_h L^2 + 16h^2\gamma^2 Md.$$

Iterating the recursion,

$$H_\pi(\rho_k) \leq e^{-\frac{3\beta}{2}hk}H_\pi(\rho_0) + \frac{24\gamma^2 h^3 \eta_h L^2}{1 - e^{-\frac{3\beta}{2}h}} + \frac{16h^2\gamma^2 Md}{1 - e^{-\frac{3\beta}{2}h}}$$
$$\leq e^{-\frac{3\beta}{2}hk}H_\pi(\rho_0) + \frac{22\gamma^2 \eta_h L^2 h^2}{\beta} + \frac{15\gamma^2 Mhd}{\beta}$$
$$\leq e^{-\frac{3\beta}{2}hk}H_\pi(\rho_0) + \frac{50hd\gamma^2(\eta_h + M)}{\beta}$$

where we used $1 - e^{-a} \geq 3a/4$ for $a \in (0, 1/4]$. Now using Lemma 6 for initialization, picking the assumed stepsize, after $k \geq \tilde{\Omega}(M\gamma^2 d/\beta^2\delta)$ iterations, we have $H_\pi(\rho_k) \leq \delta$. $\qquad\square$

## E  Proof for Section 4.4: Alternative Backward Discretization Scheme

**Lemma 8** (Implicit in Lemma 6 of [23]). *For a matrix $S = (I + t\nabla^2 f(x))^{-2}$, assuming (1) $-L \cdot I_d \preceq \nabla^2 f \preceq L \cdot I_d$; (2) $\|\nabla^2 f(x) - \nabla^2 f(y)\|_{op} \leq M\|x - y\|$ for all $x, y$; (3) $0 \leq t \leq \min\{1/8L, 1/M\}$, we have*

$$\|\nabla_k S\|_{op} := \left\|\frac{\partial S}{\partial x_k}\right\|_{op} \leq 4tM$$

*for all $k \in [d]$. Above (2) also implies $\|\nabla_i \nabla^2 f(x)\|_{op} \leq M$ for all $i \in [d]$.*

Below we give the main technical argument for this section.

**Lemma 9** (SDE Derivation). *If $t \leq \mathcal{O}(1/\gamma, 1/K)$, under Assumption 3,5,7, the backward discretization dynamics in* (18) *follows the SDE in* (10) *with*

$$\frac{4}{9}[\nabla^2\phi(X_t)]^{-1} \preceq G \preceq 4[\nabla^2\phi(X_t)]^{-1}$$

*and $\|\hat{\mu}\|^2_{\nabla^2\phi} = \mathcal{O}(t^2\gamma^2 L^2 + t^2\alpha^{-1}d^3 K^2)$. In particular, the norm decays with $t$ therefore there is no asymptotic bias with vanishing stepsize.*

*Proof.* For the process in (18), if we introduce $\tilde{Y}_t = Y_t + t\nabla f(\nabla\phi^*(Y_t))$, then we see that it evolves as a scaled Brownian motion:

$$d\tilde{Y}_t = \sqrt{2}[\nabla^2\phi^*(Y_t)]^{-1/2}dW_t = \sqrt{2\nabla^2\phi(X_t)}dW_t\,. \tag{26}$$

Therefore if we speculate that $Y_t$ takes the form of $dY_t = \mu dt + \sqrt{2G}dW_t$, then Itô's lemma gives

$$d\tilde{Y}_t = \left(\nabla f(X_t) + \left(I + t[\nabla^2\phi(X_t)]^{-1}\nabla^2 f(X_t)\right)^\top \mu + \text{Tr}(\sqrt{G}^\top T\sqrt{G})\right) dt$$
$$+ \sqrt{2}(I + t[\nabla^2\phi(X_t)]^{-1}\nabla^2 f(X_t))^\top \sqrt{G}dW_t$$

for $T = t[\nabla^2\phi(X_t)]^{-1}\frac{\partial\nabla^2 f(X_t)}{\partial Y_t} + t\frac{\partial[\nabla^2\phi(X_t)]^{-1}}{\partial Y_t}\nabla^2 f(X_t)$. Comparing this with the Brownian motion SDE for $\tilde{Y}_t$ (26), we have

$$\sqrt{G} = (I + t\nabla^2 f(X_t)[\nabla^2\phi(X_t)]^{-1})^{-1}[\nabla^2\phi(X_t)]^{1/2},$$

$$\mu = -(I + t\nabla^2 f(X_t)[\nabla^2\phi(X_t)]^{-1})^{-1}[\nabla f(X_t) + \text{Tr}(T\sqrt{G}\sqrt{G}^\top)].$$

Now to translate to the primal $X$-space through mapping $\nabla\phi^*$, another application of Itô's lemma tells us

$$dX_t = d\nabla\phi^*(Y_t) = \sqrt{2}[\nabla^2\phi(X_t)]^{-1}\sqrt{G}dW_t + \left[\text{Tr}\left(\sqrt{G}^\top\frac{\partial^2\nabla\phi^*(Y_t)}{\partial Y_t^2}\sqrt{G}\right) + [\nabla^2\phi(X_t)]^{-1}\mu\right]dt$$

$$= \sqrt{2}(\nabla^2\phi(X_t) + t\nabla^2 f(X_t))^{-1}[\nabla^2\phi(X_t)]^{1/2}\,dW_t + \left[\text{Tr}\left(\frac{\partial^2\nabla\phi^*(Y_t)}{\partial Y_t^2}\sqrt{G}\sqrt{G}^\top\right) + [\nabla^2\phi(X_t)]^{-1}\mu\right]dt$$

$$=: \sqrt{2\tilde{G}}\,dW_t + \tilde{\mu}\,dt$$

for which $\sqrt{\tilde{G}} = (\nabla^2\phi(X_t) + t\nabla^2 f(X_t))^{-1}[\nabla^2\phi(X_t)]^{1/2} \succ 0$ by the choice of $t$ and we can calculate the $\hat{\mu}$ in (10) as

$$\hat{\mu} = \tilde{\mu} - \nabla\cdot\tilde{G}(X_t) + \tilde{G}(X_t)\nabla f(X_t)$$

$$= -[\nabla^2\phi(X_t)]^{-1}\text{Tr}\left(\nabla^3\phi(X_t)[\nabla^2\phi(X_t)]^{-1}G[\nabla^2\phi(X_t)]^{-1}\right) - \nabla\cdot\tilde{G} - \sqrt{\tilde{G}}[\nabla^2\phi(X_t)]^{-1/2}\text{Tr}(TG)$$

$$- t\sqrt{\tilde{G}}[\nabla^2\phi(X_t)]^{-1/2}\nabla^2 f(X_t)\sqrt{\tilde{G}}[\nabla^2\phi(X_t)]^{-1/2}\nabla f(X_t)$$

$$= -\nabla\cdot\tilde{G} - t\sqrt{\tilde{G}}[\nabla^2\phi(X_t)]^{-1/2}\nabla^2 f(X_t)\sqrt{\tilde{G}}[\nabla^2\phi(X_t)]^{-1/2}\nabla f(X_t) - [\nabla^2\phi(X_t)]^{-1}\text{Tr}(\nabla^3\phi(X_t)\tilde{G})$$

$$- t\sqrt{\tilde{G}}[\nabla^2\phi(X_t)]^{-1/2}\text{Tr}\left(\frac{\partial\nabla^2 f(X_t)}{\partial Y_t}[\nabla^2\phi(X_t)]\tilde{G}\right)$$

$$+ t\sqrt{\tilde{G}}[\nabla^2\phi(X_t)]^{-3/2}\text{Tr}(\nabla^3\phi(X_t)[\nabla^2\phi(X_t)]^{-1}\nabla^2 f(X_t)[\nabla^2\phi(X_t)]\tilde{G}).$$

Moreover, since $0 \le t < 1/2\gamma$, under Assumption 7, we have

$$\frac{4}{9}[\nabla^2\phi(X_t)]^{-1} \preceq \tilde{G} \preceq 4[\nabla^2\phi(X_t)]^{-1}, \tag{27}$$

establishing the first claim. For the shifted drift $\|\hat{\mu}\|_{\nabla^2\phi}^2$, we look at it term by term: using Assumption 3 and (27),

$$\|t\sqrt{\tilde{G}}[\nabla^2\phi(X_t)]^{-1/2}\nabla^2 f(X_t)\sqrt{\tilde{G}}[\nabla^2\phi(X_t)]^{-1/2}\nabla f(X_t)\|_{\nabla^2\phi}^2 \le 16t^2\gamma^2\|\nabla f(X_t)\|_{[\nabla^2\phi(X_t)]^{-1}}^2$$
$$\le 16t^2\gamma^2 L^2.$$

For $\|-\nabla\cdot\tilde{G} - [\nabla^2\phi(X_t)]^{-1}\text{Tr}(\nabla^3\phi(X_t)\tilde{G})\|_{\nabla^2\phi}^2$, we have using the product rule for the divergence operator, and writing $\tilde{G}$ as $[\nabla^2\phi(X_t)]^{-1}(I + t\nabla^2 f(X_t)[\nabla^2\phi(X_t)]^{-1})^{-2}$,

$$\left\|[\nabla^2\phi(X_t)]^{-1}\text{Tr}\left(\nabla^3\phi(X_t)[\nabla^2\phi(X_t)]^{-1}(I + t\nabla^2 f(X_t)[\nabla^2\phi(X_t)]^{-1})^{-2}\right)\right.$$
$$- [\nabla^2\phi(X_t)]^{-1}\nabla\cdot(I + t\nabla^2 f(X_t)[\nabla^2\phi(X_t)]^{-1})^{-2}$$
$$\left.- [\nabla^2\phi(X_t)]^{-1}\text{Tr}\left(\nabla^3\phi(X_t)[\nabla^2\phi(X_t)]^{-1}(I + t\nabla^2 f(X_t)[\nabla^2\phi(X_t)]^{-1})^{-2}\right)\right\|_{\nabla^2\phi(X_t)}^2$$

$$= \|\nabla\cdot(I + t\nabla^2 f(X_t)[\nabla^2\phi(X_t)]^{-1})^{-2}\|_{[\nabla^2\phi(X_t)]^{-1}}^2$$

which can be bounded to scale with $t$ provided $\nabla^2 f(X_t)[\nabla^2\phi(X_t)]^{-1}$ is Lipschitz and $-\gamma I \preceq \nabla^2 f(X_t)[\nabla^2\phi(X_t)]^{-1} \preceq \gamma I$ is bounded. Using Lemma 8 with $t$ small, for $S := (I + t\nabla^2 f(X_t)[\nabla^2\phi(X_t)]^{-1})^{-2}$, we have $\|\nabla_k S\|_{op} \le 4tK$ for all $k \in [d]$ so with Assumption 5,

$$\|\nabla\cdot S\|_{[\nabla^2\phi(X_t)]^{-1}}^2 \le \alpha^{-1}\|\nabla\cdot S\|_2^2 \le \alpha^{-1}d^2\sum_{k\in[d]}\|\nabla_k S\|_{op}^2 = \mathcal{O}(\alpha^{-1}d^3 t^2 K^2).$$

For the remaining last two terms in $\hat{\mu}$ that is up to constants equal to (by Von Neumann's trace inequality and (27) above, using a similar derivation to those in A.1)

$$\left\| t[\nabla^2\phi(X_t)]^{-1}\operatorname{Tr}(\nabla^3\phi(X_t)[\nabla^2\phi(X_t)]^{-1}\nabla^2 f(X_t)) - t[\nabla^2\phi(X_t)]^{-1}\nabla\cdot\nabla^2 f(X_t)\right\|^2_{[\nabla^2\phi(X_t)]^{-1}},$$

the Lipschitz condition required for $[\nabla^2\phi(X_t)]^{-1}\nabla^2 f(X_t)$ will give a bound on this quantity as well, as it simply being the divergence of this former expression. Hence Assumption 7, together with Lemma 8 give $\|\nabla_k([\nabla^2\phi(X_t)]^{-1}\nabla^2 f(X_t))\|_{op} \leq K$ for all $k \in [d]$ and the term can be upper bounded as $t^2\alpha^{-1}\|\nabla\cdot([\nabla^2\phi(X_t)]^{-1}\nabla^2 f(X_t))\|_2^2 \leq \mathcal{O}(t^2 K^2 d^3\alpha^{-1})$. Putting things together, we have $\|\hat{\mu}\|^2_{\nabla^2\phi} \leq \mathcal{O}(t^2\gamma^2 L^2 + t^2 d^3 K^2\alpha^{-1})$ under the assumed condition in the lemma statement. $\quad\square$

The important thing to note is that the diffusion and the (shifted) drift term only involves $X_t$ and not $X_0$, which would introduce errors coming from the stochastic Brownian motion term and prevents a tighter control. Now we can essentially follow the template in Proposition 3 to finish the proof. The analysis is a somewhat tedious calculation.

*Proof of Proposition 4.* Using Mirror LSI and Lemma 9, the claim is just a stone's throw away,

$$
\begin{aligned}
\frac{d}{dt}H_\pi(\rho_t) &= \int \frac{d\rho_t}{dt}\log\frac{\rho_t}{\pi}dx \\
&= -\mathbb{E}_{\rho_t}\left[\left\|\nabla\log\frac{\rho_t}{\pi}\right\|_G^2\right] + \mathbb{E}_{\rho_{0,t}}\left[\langle\hat{\mu},\nabla\log\frac{\rho_t}{\pi}\rangle\right] \\
&\leq -\frac{4}{9}\mathbb{E}_{\rho_t}\left[\left\|\nabla\log\frac{\rho_t}{\pi}\right\|^2_{[\nabla^2\phi]^{-1}}\right] + \mathbb{E}_{\rho_{0,t}}[\|\hat{\mu}\|^2_{\nabla^2\phi}] + \frac{1}{4}\mathbb{E}_{\rho_t}\left[\left\|\nabla\log\frac{\rho_t}{\pi}\right\|^2_{[\nabla^2\phi]^{-1}}\right] \\
&\leq -\frac{\beta}{4}H_\pi(\rho_t) + \mathbb{E}_{\rho_{0,t}}[\|\hat{\mu}\|^2_{\nabla^2\phi}] \\
&\leq -\frac{\beta}{4}H_\pi(\rho_t) + C\cdot(t^2\gamma^2 L^2 + t^2\alpha^{-1}d^3 K^2).
\end{aligned}
$$

If $0 \leq t \leq h \leq \mathcal{O}(\min(1/\gamma, 1/K))$,

$$\frac{d}{dt}H_\pi(\rho_t) \leq -\frac{\beta}{4}H_\pi(\rho_t) + h^2 C(\gamma^2 L^2 + \alpha^{-1}d^3 K^2).$$

Written differently,

$$\frac{d}{dt}\left(e^{\frac{\beta}{4}t}H_\pi(\rho_t)\right) \leq e^{\frac{\beta}{4}t}h^2 C(\gamma^2 L^2 + \alpha^{-1}d^3 K^2).$$

Integrate it for $0 \leq t \leq h$, we have for $h \leq \frac{1}{\beta}$,

$$
\begin{aligned}
e^{\frac{\beta}{4}h}H_\pi(\rho_h) - H_\pi(\rho_0) &\leq \frac{4}{\beta}(e^{\frac{\beta h}{4}} - 1)h^2 C(\gamma^2 L^2 + \alpha^{-1}d^3 K^2) \\
&\leq 2h^3 C(\gamma^2 L^2 + \alpha^{-1}d^3 K^2)
\end{aligned}
$$

where we used $e^a \leq 1 + 2a$ for $a \in (0,1]$. Altogether this gives us

$$
\begin{aligned}
H_\pi(\rho_h) &\leq e^{-\frac{\beta}{4}h}H_\pi(\rho_0) + e^{-\frac{\beta}{4}h}2h^3 C(\gamma^2 L^2 + \alpha^{-1}d^3 K^2) \\
&\leq e^{-\frac{\beta}{4}h}H_\pi(\rho_0) + 2h^3 C(\gamma^2 L^2 + \alpha^{-1}d^3 K^2).
\end{aligned}
$$

Iterating the recursion,

$$
\begin{aligned}
H_\pi(\rho_k) &\leq e^{-\frac{\beta}{4}hk}H_\pi(\rho_0) + \frac{2h^3 C(\gamma^2 L^2 + \alpha^{-1}d^3 K^2)}{1 - e^{-\frac{\beta}{4}h}} \\
&\leq e^{-\frac{\beta}{4}hk}H_\pi(\rho_0) + \frac{12h^2 C(\gamma^2 L^2 + \alpha^{-1}d^3 K^2)}{\beta}
\end{aligned}
$$

where we used $1 - e^{-a} \geq 3a/4$ for $a \in (0, 1/4]$. Now using Lemma 6 for initialization, picking the assumed stepsize, after $k \geq \tilde{\Omega}(\sqrt{\gamma^2 L^2 + \alpha^{-1}d^3 K^2}/\delta^{1/2}\beta^{3/2})$ iterations, we have $H_\pi(\rho_k) \leq \delta$. $\quad\square$