# OpenReview forum: "Mirror Langevin Monte Carlo: the Case Under Isoperimetry"
_NeurIPS.cc/2021/Conference — NeurIPS 2021 Poster_

### Official Review · Reviewer_T7ux · 2021-06-25

**Rating:** 7
**Confidence:** 5

**Summary:**

This paper gives a discretization analysis of the mirror-Langevin Monte Carlo algorithm, which is the analogue of mirror descent for MCMC. The prior works studying discretization of mirror-Langevin are as follows:

[1] Zhang et al., who studied the Euler-Maruyama discretization. They used stochastic calculus coupling techniques in the vein of works by e.g. Dalalyan.

[2] Ahn and Chewi, who introduced and studied an alternate discretization which simulates the diffusion step more faithfully. Their analysis uses Wasserstein calculus in the vein of Durmus, Majewski, and Miasojedow.

A key difference between these two works is that the result of [1] has a non-vanishing bias (even with vanishing step size), whereas the discretization of [2] does not.

This paper studies both the Euler-Maruyama and Ahn-Chewi discretizations, and introduces a third one which combines the Ahn-Chewi discretization with a proximal step to obtain better guarantees. The results obtained have non-vanishing bias for the Euler-Maruyama discretization, and vanishing bias for the latter two discretizations, which is in line with previous work. The techniques are inspired by the discretization analyses of Wibisono, who is inspired by the interpretation of sampling as composite optimization in Wasserstein space. Moreover, this paper emphasizes the use of a mirror-log-Sobolev inequality as the condition ensuring rates of convergence, which allows for treating examples of non-convex potentials (which are not handled by the prior two works).

The main contributions can be summarized thus: (1) by obtaining a similar dichotomy between the bias of Euler-Maruyama vs. Ahn-Chewi, it sheds further light on the effect of discretization of the diffusion in the presence of non-isotropic diffusions; (2) it shows how to adapt a third type of discretization analysis to the mirror-Langevin setting; (3) it introduces a proximal mirror-Langevin algorithm which had previously not been considered and gives guarantees.

**Limitations And Societal Impact:**

Yes.

**Main Review:**

Evaluation: This paper is a good contribution to the sampling literature in light of the main contributions listed above. Since it is more challenging to handle the non-isotropic diffusion than the study of vanilla Langevin, understanding discretization analysis for mirror-Langevin is an important problem.

My main concern with this work is that it is not straightforward to find examples in which the mirror log-Sobolev inequality holds (besides when the mirror map is quadratic, in which case it reduces to the usual log-Sobolev inequality). For example, the usual log-Sobolev inequality is implied by strong convexity, and besides there are many identified situations where it holds (e.g. convolution of a measure with bounded support with a Gaussian). On the other hand, relative strong convexity w.r.t. a mirror map is not enough to imply a mirror LSI; for instance, as discussed in the paper of Chewi et al. on mirror-Langevin, when the mirror map equals the potential then relative strong convexity trivially holds, but a mirror LSI does not hold without further assumption on the potential/mirror map (see the reference to Bobkov-Ledoux therein). It is important therefore to find further examples in which the mirror LSI holds to assess whether this is indeed a useful assumption.

Specific Comments:
- On pg. 3, after Assumption 2, it is claimed that the assumption of a mirror LSI is “strictly weaker” than that of relative strong convexity, but this is false as discussed above. Rather, it would be better to include somewhere in the paper a more thorough comparison of the two sets of assumptions.
- Last display equation on pg. 4 should have \nabla_x rather than \nabla x.
- On pg. 6, it is claimed that “the simplest EM discretization exhibit [sic] an irreducible bias”. It is not clear what exactly is meant by this statement: after all, Thm. 1 (as well as the prior work by Zhang et al.) merely provide upper bounds on the bias, and it is not yet proven if the EM discretization indeed exhibits bias or not.
- On the subject of the bias of EM discretization: the O(d^2) bias is quite large. For comparison purposes, what is the size of the relative entropy at initialization under the same assumptions, for a sensible choice of initialization (e.g. Gaussian with appropriate variance)? Is it clear that this bias term O(d^2) actually constitutes an improvement over the initialization? If not, then the EM result seems almost vacuous to me.
- Lemma 6 in the appendix is surely wrong; the relative entropy of ρ_0 w.r.t. to π is only finite if ρ_0 is absolutely continuous w.r.t. π, and this cannot hold if ρ_0 is taken to be a Dirac mass. It means that Thm. 1 which initializes (4) at a Dirac mass is incorrect as stated (or at least the proof has to be modified to show that the relative entropy becomes finite after finitely many iterations).
- Throughout the paper there are a number of minor grammatical errors, and the paper would benefit from an extra round of proofreading.

**Time Spent Reviewing:**

1

---

> ### Author Response · Authors · 2021-08-09
> **Response #4**
>
> Thank you for all the thoughtful comments and suggestions.
>
> - “Assumption”: Yes it should be clarified that relative strong convexity isn’t enough to imply Mirror-LSI without further assumptions. Although there exists sufficient conditions e.g., [Bobkov, Ledoux, “From Brunn–Minkowski to Brascamp–Lieb and to logarithmic Sobolev inequalities”] Page 3 bottom for examples where Mirror-LSI holds for a general dual norm; other examples appear on e.g., Page 13. We plan on giving more examples and including more discussions on the related literature in the revised version.
> - “Initialization in Lemma 6”: Thanks for pointing it out -- yes the initialization should have read as the normal distribution centered at the stationary point instead of the Dirac mass and the KL divergence scales as O(d); although the dependence on the dimension isn’t the whole story here, as the dependence on $\zeta$, the self concordance parameter, really captures the scaling of the bias. In the case when $\phi=1/2\|x\|^2$, one would get $\zeta=0$ and of course vanishing bias with stepsize; and therefore there’s a sense in which the geometry can only change so fast for the EM discretization to make sense. But again, we agree that as future steps, a tight characterization/lower bound on the bias would shed more light in comparing these schemes.

---

> > ### Comment · Reviewer_T7ux · 2021-08-09
> > **Response**
> >
> > Thank you for your response. Regarding the mirror LSI, the Bobkov-Ledoux result is not convincing to me -- in fact, quite the opposite, because it indicates that a mirror LSI assumption may be quite restrictive and special (their result only applies to the case when the mirror map equals the potential, and moreover when the potential satisfies a strong condition; it is even less clear how a mirror LSI might hold when the mirror map and potential are substantially different). I understand that obtaining further examples of when the mirror LSI holds is a separate research question in itself. However, I am still curious to hear if, in the opinion of the author(s), whether the mirror LSI (for non-quadratic mirror maps) is likely to be substantially more useful than the relative convexity assumption; the latter assumption at least has the benefit of a number of works in the optimization literature which give examples of when it holds.

---

> > > ### Author Response · Authors · 2021-08-10
> > > **Follow-up**
> > >
> > > Thanks for the follow-up.
> > >
> > > Looking at the mirror-LSI condition $Ent_\pi(g^2) \leq c \int \|\nabla g\|_{[\nabla^2 \phi(x)]^{-1}}^2 d\pi$ for any bounded smooth function $g$, if we pick $\phi(x)$ to be smooth on the domain of $f$ modulo a set of measure zero (doesn’t have to be quadratic), then just by appealing to the classical LSI we know there are non-strongly-convex potential for which the condition holds. There is also related work from [Patrick Cattiaux, Arnaud Guillin & Li-Ming Wu, “Some Remarks on Weighted Logarithmic Sobolev Inequality”], but there are caveats there for us to directly apply their technique to our setting. But these are really just observations - working on a general characterization for such a condition to hold is part of an ongoing research and would likely be an interesting result/paper on its own.
> > >
> > > We should also add that, most of the analysis carried out here could be generalized to say, a mirror version of Poincare inequality and/or with different divergence metrics in a more or less straightforward way if one wishes to work with those.

---

### Official Review · Reviewer_82Yq · 2021-07-14

**Rating:** 5
**Confidence:** 4

**Summary:**

The paper under review mainly studies different discretization schemes for mirror Langevin dynamics. The main assumption is the non-asymptotic convergence/long time convergence behavior to the target invariant measure. The main feature of this type Langevin dynamic is that they have non-gradient drift and variable diffusion matrix. The author try to covey the importance of the underlying geometry which is introduced by the variable diffusion matrix.

**Limitations And Societal Impact:**

Negative.

1. The Mirror Langevin dynamics presented in this paper is a special case of a series paper for general Langevin dynamics.  e.g. [Q. Feng and W. Li] Hypoelliptic entropy dissipation for stochastic differential equations. The dynamics (3) is simply by choosing $a(X_t)=[\nabla^2\phi(X_t)]^{-1}$ and $\gamma=0$ for the dynamics (1) in  [Q. Feng and W. Li].  The Fokker-Planck equation is also derived there together with the entropy decay result Proposition 1. (Note here the LSI is assumed to be true in the current paper, thus the decay of KL seems to follow naturally.)

2. The assumption of the current paper claims to be weaker comparing to the existing ones [23]. The reviewer is quite curious about the verification of Assumption 2. In the aforementioned paper [Q. Feng and W. Li], the existence of the LSI for a general variable diffusion Langevin dynamic follows from the Ricci curvature lower bound. Can the author verify that Assumption 2 is indeed true for certain classes of the mirror map $\phi$. It would very interesting to see such examples where global in time and space LSI is established for non-convex f.



**Main Review:**

Positive.

1. The logic of the paper is easy to follow, and the several discretization schemes are presented with complete derivation.

2. The idea of introducing new  discretization schemes seems to be valid for more general variable diffusion Langevin dynamics.


**Time Spent Reviewing:**

8

---

> ### Author Response · Authors · 2021-08-09
> **Response #3**
>
> - “Related work”: Thank you for bringing up the reference and the comments. We note that while assumption 2 and the continuous dynamics in our work bear resemblance to the one considered in [Feng, Li ‘21], the focus of the paper is really on studying the (implementable) discretized counterpart using different numerical schemes under various weaker regularity conditions, which is not present in the above work. And indeed this forms the bulk of our algorithm analysis. In this regard, we do believe the emphasis of the two papers are different.
> - “Assumption”: Mirror LSI, being a strict generalization of the more classical LSI by taking $\phi = 1/2\|x\|^2$, does cover certain non-convex potentials $f$, which is excluded in [23] by simply observing that their assumption implies $\nabla^2f \succeq \mu \nabla^2 \phi \succ 0$. Additionally, in the case $\phi=f$ (i.e., Newton Langevin), there exists condition for checking such an assumption (this is further elaborated in [Bobkov, Ledoux, “From Brunn–Minkowski to Brascamp–Lieb and to logarithmic Sobolev inequalities”]).

---

### Official Review · Reviewer_dzsF · 2021-07-14

**Rating:** 6
**Confidence:** 4

**Summary:**

This paper studies discretized mirror Langevin diffusions under the assumption of mirror-log Sobolev inequalities as well as relative smoothness assumptions. In particular, they show that a new backward method obtains perhaps the best dependence in terms of finite sample guarantees in this setting. On the other hand, Euler-Maruyama discretization has an irreducible bias and a different forward discretization has an exponential dependence on the diameter of the mirror set.

**Limitations And Societal Impact:**

Limitations are not explicitly discussed. In particular, the analysis done is all in terms of upper bounds, which may not be optimal. Furthermore, the performance of the forward and backward schemes are not thoroughly explored in practice, and in particular, the differences in solving (12) versus (16).

**Main Review:**

Upsides:

+ This paper gives new upper bounds for discrete mirror Langevin dynamics.
+ The upper bounds on these three different discretizations of mirror Langevin dynamics allow one to compare them theoretically.
+ The work proposes a new backward discretization that has the best guarantees.
+ This work tackles the analysis of an important generalization of Langevin dynamics, that may hold the key to more efficient sampling algorithms.
+ The paper is generally well written and highlights the differences in the upper bounds obtained.
+ The assumption of weaker relative smoothness is interesting.

Downsides:

- The methodology and assumptions have mostly been used before, with the exception of the weaker relative smoothness and the new discretization.
- Proposition 1 appears to be a restatement of in Chewi et al. 2020. Theorem 1. (since Poincare is weaker than log Sobolev *edit due to typo)
 - Experiments do not enlighten differences between the forward discretization of Ahn and Chewi versus the new backward discretization. For example, the difference in dependence on $M$ for the forward versus backward schemes is in terms of upper bounds rather than lower bounds, and it would be nice to see experiments that illustrate this difference. Also, it is unclear how different (12) versus (16) is in terms of hardness to solve. (*edit: this should be (11) versus (17), as the authors point out)
- It is unclear how tight the analysis in Theorem 1 and Proposition 3 are. In particular, does Theorem 1 necessarily imply that the bias is non-vanishing? Are the bounds in Proposition 3 in some sense tight?
- The performance of EM in Figure 1 is strange in light of Figure 3 in Chewi et al. 2020. What is the difference in implementation here?
- The comment at the bottom of page 8 is not clear. Why is mini batching equivalent to adding Gaussian noise?

(edit fixed a few typos)

---

After considering the authors' response and discussion with the other reviewers, I am raising my score to 6 and recommending acceptance based on the theoretical merit of the paper, although I do not give a higher score due to the weaker practical details.

**Time Spent Reviewing:**

2

---

> ### Author Response · Authors · 2021-08-09
> **Response #2**
>
> We appreciate all the comments and questions.
>
> - “Methodology”: As nicely summarized by R4, the existing analysis for the discretized algorithms typically rely on coupling techniques or draw on tools from convex optimization, whereas the analysis here requires quite different ingredients for controlling the discretization error, and the weaker notion of smoothness/convexity assumptions introduce difficulties for obtaining meaningful convergence guarantees. (i.e., we don't share much similarity with Ahn & Chewi and Zhang et al. when it comes to analysis)
> - “Proposition 1”: As alluded to by other reviewers, there are different ways to go about showing exponential convergence for the continuous dynamics. Chewi et al. worked with $\chi^2$ divergence, which plays nicely with the Poincare inequality under their consideration, and translated back the result to other information divergence. We took a more direct approach for calculating the time derivative of KL divergence, which is a more natural potential to keep track of for Log-Sobolev inequality, and it remains to note that the implication of Log-Sobolev for Poincare is also not immediate under general metric tensor.
> - “Hardness of (12) and (16)”: They are the same, both approximately solved by EM (in experiments, we used 10 steps). Perhaps you meant to refer to (11) and (17), which is the main difference between backward and forward. In the case when there’s closed-form expression for $\nabla \phi^*$, (11) is a linear-time O(d) operation, whereas (17) involves solving a convex optimization problem. In the case when the inversion $\nabla \phi^*$ has to be done numerically, both would amount to solving a convex optimization problem. In simulations, these were performed by simple gradient steps.
> - “Forward vs. Backward”: For the few examples that we have experimented with (some of which not included in the draft), it’s hard to tell if there are significant differences in practice in terms of forward vs. backward discretization. It is conceivable that one might be able to tighten the analysis for forward discretization somewhat (either weaker assumption or improved rate); however, we’d like to note that even in the vanilla Langevin case, there is a gap between the best known rates for the proximal and gradient-based discretization, without additional stringent assumption such as dissipativity (and again it’s an intriguing open question whether the gap is real or not).
> - “Tightness”: We agree it would be very interesting to probe the limit of these schemes and it is indeed a major open question to establish and formalize a general framework for studying lower bounds for MCMC algorithms. However, it is our intention for this paper to focus on upper bounds and study the best rate one could get under the weakest set of assumptions for Langevin algorithm working under general geometries. In terms of bias, qualitatively as shown in the experiment, there does appear to be a difference when one discretizes the geometry $\phi$, which is in line with what the theory suggests.
> - “Figure 1”: The stepsize choice and #steps ran are different for the two experiments. Our discretized EM implementation of NLA used gradient steps for numerical inversion, checking for stopping criteria when the gradient norm is $<10^{-4}$; Chewi et al. focused mostly on continuous dynamics, and the exposition in the text wasn’t detailed enough for us to glean what specific discretization/inner solver was employed for producing the NLA part of Figure 3. But our reproducible code is included as part of the submission.

---

### Official Review · Reviewer_MHDF · 2021-07-20

**Rating:** 7
**Confidence:** 3

**Summary:**

In this work the authors study the convergence of mirror descent langevin dynamics -- Langevin dynamics which have been transformed appropriately to satisfy certain constraints or evolve according to a specific non-euclidean geometry.   The dynamics can be characterised in various forms, including as a riemannian manifold overdamped langevin process with a specific choice of metric tensor -- but also as an SDE in the dual (unconstrained space) which is projected onto the original state space.

The paper considers the convergence to equilibrium of the mirror descent Langevin dynamics -- both for the original continuous time process, but also for various discretisations of the scheme.  This is done under the assumption of a Logarithmic Sobolev Inequality, thus generalising common convergence analysis which typically require strong-log-concavity assumptions.

It is demonstrated that the choice of discretisation plays a key role in achieving consistency under vanishing step-size for constraints which involve non-smooth contributions (e.g. constraint to lie within a box, etc), ultimately presenting a backward discretisation scheme which is far more stable to sudden changes in the geometry underlying the constraint.

These results are illustrated through a straightforward numerical example.





**Ethical Concerns:**

no concerns

**Limitations And Societal Impact:**

Limitations have been adequately addressed -- societal impacts have not been mentioned, though I certainly cannot envisage any.

**Main Review:**

I found the paper generally well written, to the best of my ability in understanding the details.  The general arguments are familiar from other works, but this pulls them together to provide analysis for an important problem.  Based on this I believe the work demonstrates significance, particularly in identifying the role of the smoothness of the geometric induced by the constraint.

The main issue I find with this work is the lack of convincing numerical examples to demonstrate this -- the single presented example is important as a toy illustrative example, but is trivial in many senses.  Perhaps one could find an opportunity to demonstrate more complex forms of constraints, e.g. e.g. constraints to simplex and another geometric constraints.  What about employing this methodology simply for accelerating unconstrained sampling using Newton-type updates based on the hessian of the objective?   These could've been addressed as  examples.

Minor comments:

l14: shapes

l20: isoperimetric

l26: continuous

l35: legendre has an accent

l53:'It is evident (and reminiscent of the classical Langevin algorithm) that the discretized algorithm will converge to a biased limit' --> there are many well-known examples where the EM discretisation of a Langevin SDE will not even have a long-time limit.  So without further assumptions, it's an open question whether a limit even exists, biased or not.

l99: is the Hessian stability condition correct -- what are $x$ and $x'$ here?

l105: What is $|| \cdot ||_{[\nabla^2 \phi(x)]^{-1}}$ ?  Has this been defined?

I also highlight some important works which don't appear to have been cited/considered in this work

Wang, Xiao, Qi Lei, and Ioannis Panageas. "Fast Convergence of Langevin Dynamics on Manifold: Geodesics meet Log-Sobolev." arXiv preprint arXiv:2010.05263 (2020) which really follows the approach of Vempala & Wibiscono 2019,

Lamperski, Andrew. "Projected Stochastic Gradient Langevin Algorithms for Constrained Sampling and Non-Convex Learning." arXiv preprint arXiv:2012.12137 (2020).

**Time Spent Reviewing:**

3

---

> ### Author Response · Authors · 2021-08-09
> **Response #1**
>
> We appreciate all the comments.
>
> - “Numerical Example”: The second example in the experiments section (L323-327) gives an example where we compare ULA and Newton-Langevin for an ill-conditioned potential: $f =(x-\mu)^{\top}\Sigma^{-1}(x-\mu)$, where we show that taking $\phi=f$ (i.e., leveraging the Hessian information) instead of $\phi=1/2\|x\|^2$ significantly speeds up the convergence. Thanks for the suggestion and we plan to include more numerical examples in the revised version.
>
> - L53: Yes thanks for pointing it out - we certainly forgot the quantifiers there.
>
> - L99: The assumption of self-concordance in Assumption 1 implies that the function locally behaves like a quadratic, therefore for any two points x and x’ close to each other (in local norm), the hessian is stable up to a multiplicative factor as written.
>
> - L105: This is the local norm $\|x\|_{[\nabla^2\phi(x)]^{-1}}^2 = x^{\top}[\nabla^2\phi(x)]^{-1} x$, also briefly mentioned in L86.
>
> - “Additional References”: Thank you for the references, although in both works, the numerical algorithms under study are different from the ones we consider here (see e.g., GLA in eqn (1) & (2) for the former and projected gradient method in eqn (1) for the latter), and therefore the focus are somewhat different.

---

### Decision · Program_Chairs · 2021-09-27

**Decision:**

Accept (Poster)

**Comment:**

The paper studies discretization schemes for the mirror-descent version of Langevin dynamics, under a mirror log-Sobolev inequality (LSI). We thank the authors for a lively discussion. We have decided to recommend acceptance, because the theoretical contribution is of interest to the community. That being said, we still had concerns about 1) knowing when mirror LSI is likely to hold, and 2) the thinness of the experimental section. We thus encourage the authors to update the paper before the final version, so as to include:
* a paragraph on when mirror LSI holds, or might hold, following the discussion that was held during the reviewing process.
* More quantitative experiments that illustrate the theoretical results on the various discretization schemes. Please also make the figure labels more readable.